# Role of vector phenotypic plasticity in disease transmission as illustrated by the spread of dengue virus by *Aedes albopictus*

Dominic P. Brass [1,2] ✉, Christina A. Cobbold[3], Bethan V. Purse [1], David A. Ewing [4], Amanda Callaghan[2] & Steven M. White [1]

The incidence of vector-borne disease is on the rise globally, with burdens increasing in endemic countries and outbreaks occurring in new locations. Effective mitigation and intervention strategies require models that accurately predict both spatial and temporal changes in disease dynamics, but this remains challenging due to the complex and interactive relationships between environmental variation and the vector traits that govern the transmission of vector-borne diseases. Predictions of disease risk in the literature typically assume that vector traits vary instantaneously and independently of population density, and therefore do not capture the delayed response of these same traits to past biotic and abiotic environments. We argue here that to produce accurate predictions of disease risk it is necessary to account for environmentally driven and delayed instances of phenotypic plasticity. To show this, we develop a stage and phenotypically structured model for the invasive mosquito vector, *Aedes albopictus*, and dengue, the second most prevalent human vector-borne disease worldwide. We find that environmental variation drives a dynamic phenotypic structure in the mosquito population, which accurately predicts global patterns of mosquito trait-abundance dynamics. In turn, this interacts with disease transmission to capture historic dengue outbreaks. By comparing the model to a suite of simpler models, we reveal that it is the delayed phenotypic structure that is critical for accurate prediction. Consequently, the incorporation of vector trait relationships into transmission models is critical to improvement of early warning systems that inform mitigation and control strategies.

Vector-borne diseases (VBDs) are primarily vectored by ectothermic arthropods, whose life history is sensitive to environmental variation[1]. Our ability to predict if vector populations can sustain pathogen transmission across the species range requires an understanding of how environmental variation and vector life-history interact[2,3]. There is now a rich literature exploring the mechanisms through which environmental variation alters vector trait expression, but explicit and delayed mechanisms of individual variation are generally omitted, even in extensively studied systems[4]. For example, adult mosquitoes experiencing hot, dry summer conditions have a shorter lifespan than adults that are subject to more favourable temperatures[5]. Short-lived mosquitoes are less likely to survive to complete the extrinsic

[1]UK Centre for Ecology & Hydrology, Benson Lane, Wallingford, Oxfordshire, UK. [2]Ecology and Evolutionary Biology, School of Biological Sciences, University of Reading, Reading, UK. [3]School of Mathematics and Statistics, College of Science and Engineering, University of Glasgow, Glasgow, UK. [4]Biomathematics and Statistics Scotland, Edinburgh, UK. ✉e-mail: dombra@ceh.ac.uk

incubation period and take a subsequent blood meal to transmit the pathogen. Therefore, there is environmentally driven inter-annual and inter-regional variation in the ability of mosquitoes to vector disease[6].

Predictions of the relative risk of VBDs, via metrics such as the basic reproduction number, are highly sensitive to traits such as adult longevity. Seasonal and regional variation in this trait is often accounted for by parameterising functions that directly relate the current temperature to the longevity of adult mosquitoes[7,8]. However, adult longevity has also been shown to vary in response to an individual's experience of competition and temperature during development. Therefore, the traits expressed by individuals in the population result from complex interactions between past environmental conditions and population states[9]. By ignoring the role that a vector's historic environmental experiences have on biological traits critical to disease transmission we implicitly make the mean-field assumption, that all individuals are equally competent vectors regardless of their past experiences of the biotic and abiotic environment[10]. This mean-field assumption compromises our ability to assess the risk of vector-borne disease in variable environments, where seasonal patterns in environmental variables induce seasonal variation in vector population dynamics. Similarly, mean-field assumptions limit our ability to assess the relative risk of VBDs between climatic zones, such as between tropical and temperate environments, where different patterns of environmental variation occur.

The delayed effect of developmental experience on adult traits is an example of phenotypic plasticity, the ability of organisms expressing the same genotype to express different phenotypes according to the environmental conditions they are subject to[11]. To demonstrate how mechanisms of delayed phenotypic plasticity can be incorporated into predictions of VBDs we consider the ability of the invasive mosquito species *Aedes albopictus* to vector dengue virus. We use a recently derived phenotypically explicit mathematical modelling framework to represent how historical biotic and abiotic environmental conditions determine the trait dynamics of vector populations[12]. *Ae. albopictus* is a competent vector of dengue virus that is now widely distributed in temperate zones into which dengue is regularly imported[13]. Dengue is a viral VBD that has seen a recent dramatic increase in cases: in 2000 there were about half a million cases, which has risen to over 4.2 million in 2022, caused by a combination of climate and anthropogenic change[14].

Despite predictions of broad suitability for the transmission of dengue virus by *Ae. albopictus* in temperate climates such as Europe by both statistical and mechanistic modelling approaches, outbreaks in these regions have so far been limited in size and duration[15,16]. This mismatch between observed and predicted disease incidence motivates the development of modelling approaches that are better able to reflect the currently observed global incidence of dengue. Due to the species' global range (currently between 0 and 52.5 degrees latitude[13]), *Ae. albopictus* populations are subject to a diverse range of environmental conditions. Environmental variation has been shown to induce variable trait dynamics in field populations and this may alter the ability of populations in different environments to vector disease[17,18]. It is our hypothesis that, by accounting for the effect of biotic and abiotic environmental variation on the phenotypic trait structure and population dynamics of *Ae. albopictus*, both the population dynamics of the species across its range and the current patterns of disease incidence around the globe can be better understood. Further, we propose that omitting mechanisms of phenotypic plasticity limits the ability of models to produce accurate and generalisable predictions of disease incidence.

## Results

### Model overview and validation

We develop a Susceptible-Exposed-Infected-Resistant (SEIR) model for the transmission of dengue virus by *Ae. albopictus* that incorporates a system of environmentally driven stage and phenotypically structured delay-differential equations. The model explicitly predicts mosquito population and trait dynamics. This model can represent the instantaneous effects of the current environment (e.g., temperature, precipitation, evaporation, photoperiod, and larval density) on vector traits such as development rate and through-stage survival. In addition, it captures the delayed effect of historic biotic and abiotic environments on these same traits. This allows us to represent how phenotypic plasticity in response to historic biotic and abiotic environmental variation during development alters adult longevity. We explicitly track the number of adult mosquitoes within the population that have experienced specific developmental conditions. This results in a phenotypically structured adult population that directly accounts for the effects of past environments on the ability of mosquitoes to transmit VBDs.

We use this model to consider the infection dynamics of a dengue outbreak, beginning with the introduction of humans, infected with a single serotype of dengue virus, into a completely susceptible population. This simplified representation of the urban dengue virus transmission cycle is most applicable to non-endemic temperate and sub-tropical regions, where the likelihood of multiple serotypes circulating simultaneously is reduced and the population's prior exposure to dengue virus is limited[19]. Our model incorporates detailed species-specific processes, such as diapause, quiescence, the water dynamics of developmental habitats, and developmental plasticity that causes adult traits to vary in response to the environmental conditions experienced by juveniles throughout development[20,21]. We use data from published laboratory experiments to parametrise multi-dimensional reaction norms describing how temperature, larval density, precipitation dynamics, and photoperiod alter the traits of mosquitoes within and across developmental stages. Remotely sensed and climate reanalysis data combined with human population density data are then used to make spatio-temporal predictions of vector dynamics and VBD risk across the globe. A full description of the model is provided in the Methods section and a detailed example of the model outputs can be found in Supplementary Note 1.

We extensively validate the model predictions of mosquito population, wing length, and disease dynamics by comparing them to field surveys of mosquito populations and disease outbreaks across the species range. To validate the population dynamics we use published surveillance data, including life-stage specific population density estimates and average trait data, from 40 locations, across 13 countries and 4 continents (see Supplementary Note 1 for the full set of validations). To validate the model predictions of dengue dynamics we compare them to historical dengue outbreaks, using reports from the outbreak to select a likely introduction scenario for dengue virus into the region and to define an area over which to simulate the model. Validation data was obtained from a comprehensive search of published studies that observed the population dynamics of *Ae. albopictus* in the field or reported human cases of dengue. The literature search was conducted using a snowball procedure - a systematic search revealed a high number of false positives. Studies were included in our validation if the data was collected with at least a monthly resolution from a region where *Aedes aegypti*, a closely related vector species was absent. This exclusion criteria is necessary as *Ae. aegypti* is known to compete with *Ae. albopictus* for larval resources and is also a vector of dengue virus, factors which we do not account for here. Since substantial variation in case reporting effort through space and time is likely, our predictions of dengue transmission are not rescaled and the validations are intended to demonstrate that the model produces plausible disease dynamics (see also Supplementary Note 2). The comparisons presented in this paper represent the full set of datasets meeting the above criteria that were found in our literature search.

Across the species global range, the model achieves excellent predictions of the population and trait dynamics of *Ae. albopictus*

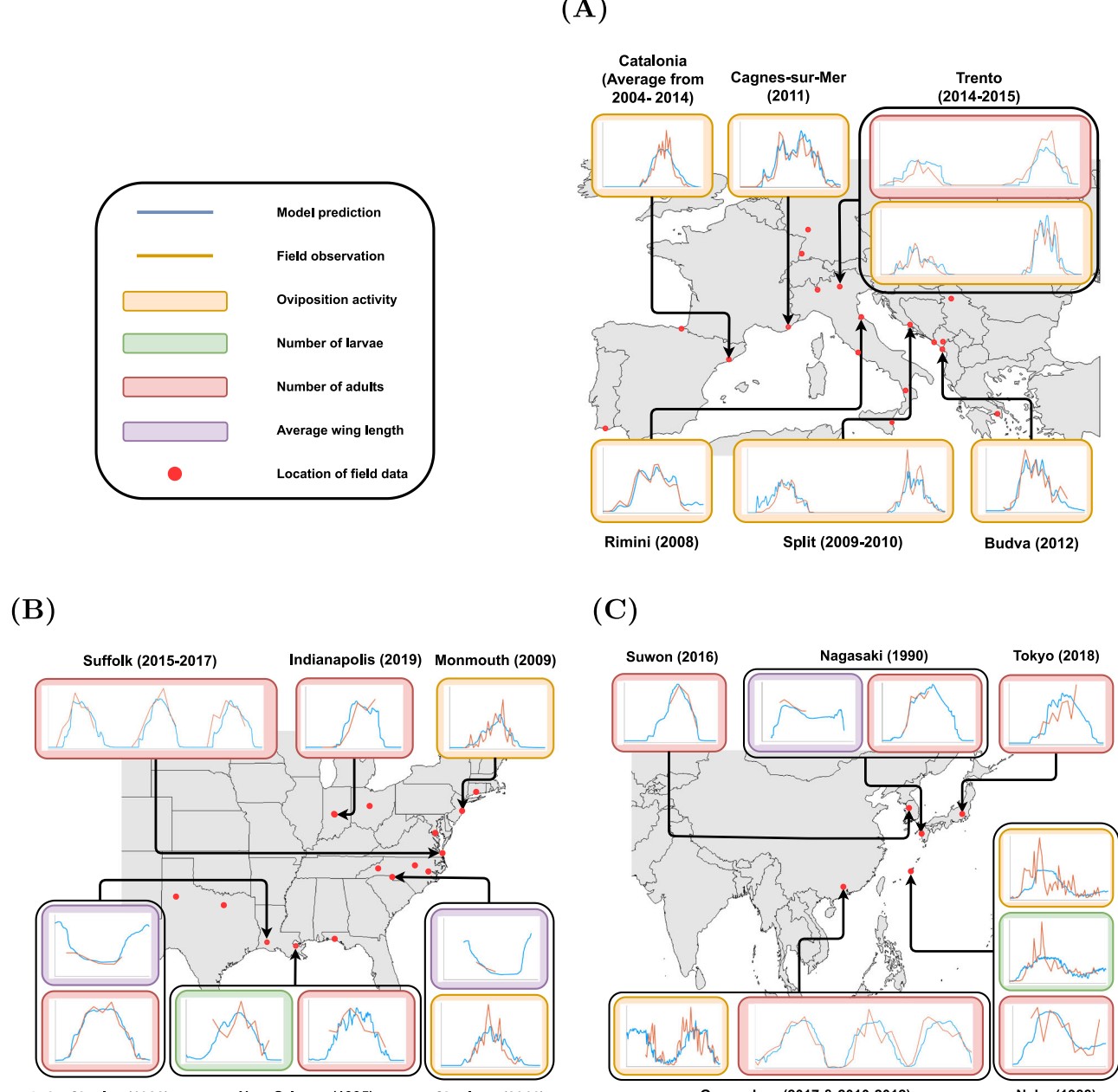

**Fig. 1 | Validation of predicted population dynamics against field data.** Validations of the scaled model predictions against field data for locations around the world. In each case, the *x*-axis is time in days, and the *y*-axis is either the scaled abundance of individuals in a specific life stage or an average trait value. Each blue line represents the model's prediction of the population dynamics at the corresponding location, the orange lines represent field observations from the same location. The colour of the outer box indicates the type of data that is being compared, orange boxes are for oviposition activity, which is the number of eggs predicted to present in an ovitrap, green boxes indicate larval numbers, red boxes indicate adult numbers and purple for average wing length. The location and year of each comparison are indicated below/above each graph, further details of each comparison can be found in Supplementary Note 1. Source data are provided as a Source Data file. **A** Comparisons for Europe[24–26,77,159–161]. **B** Comparisons for North America[17,22,162–165]. **C** Comparisons for Asia[18,27,85,166–168].

(Fig. 1). The model's predictions hold over a broad range of climatic regions and reflect the differences in population dynamics that are observed between temperate and tropical environments (see Monmouth, New Jersey, USA (2009), $R^2 = 0.59$, and St. Paul, La Réunion (2013), $R^2 = 0.65$ (Supplementary Fig. 47A)[22,23]. The model accurately predicts detailed within-season dynamics and consistently captures the species' phenology (see adult numbers in Guangzhou, China (2006–2015), $R^2 = 0.6$ (Supplementary Fig. 45B), and Rimini, Italy (2008), $R^2 = 0.94$ (Supplementary Fig. 9A)[24,25]. The model additionally captures the inter-annual variation in abundance observed in multi-year data sets (see oviposition activity in Trento, Italy (2010–2020),

$R^2 = 0.71$ (Supplementary Fig. 14), and adult numbers in Suffolk, USA (2009–2018), $R^2 = 0.64$ (Supplementary Fig. 40))[26,27].

Our model predictions indicate and confirm the current limited ability of *Ae. albopictus* to sustain autochthonous transmission in the Alpes-Maritimes Department of France. Predictions show good resemblance to the disease dynamics observed during the multi-year epidemic on the island of Réunion and the large outbreak that occurred in Guangzhou, China (Fig. 2)[28–33]. The reduced resemblance of our predictions to the observed disease dynamics towards the latter half of each outbreak may be explained by the intensification of vector control activities which we do not account for here. For example, in

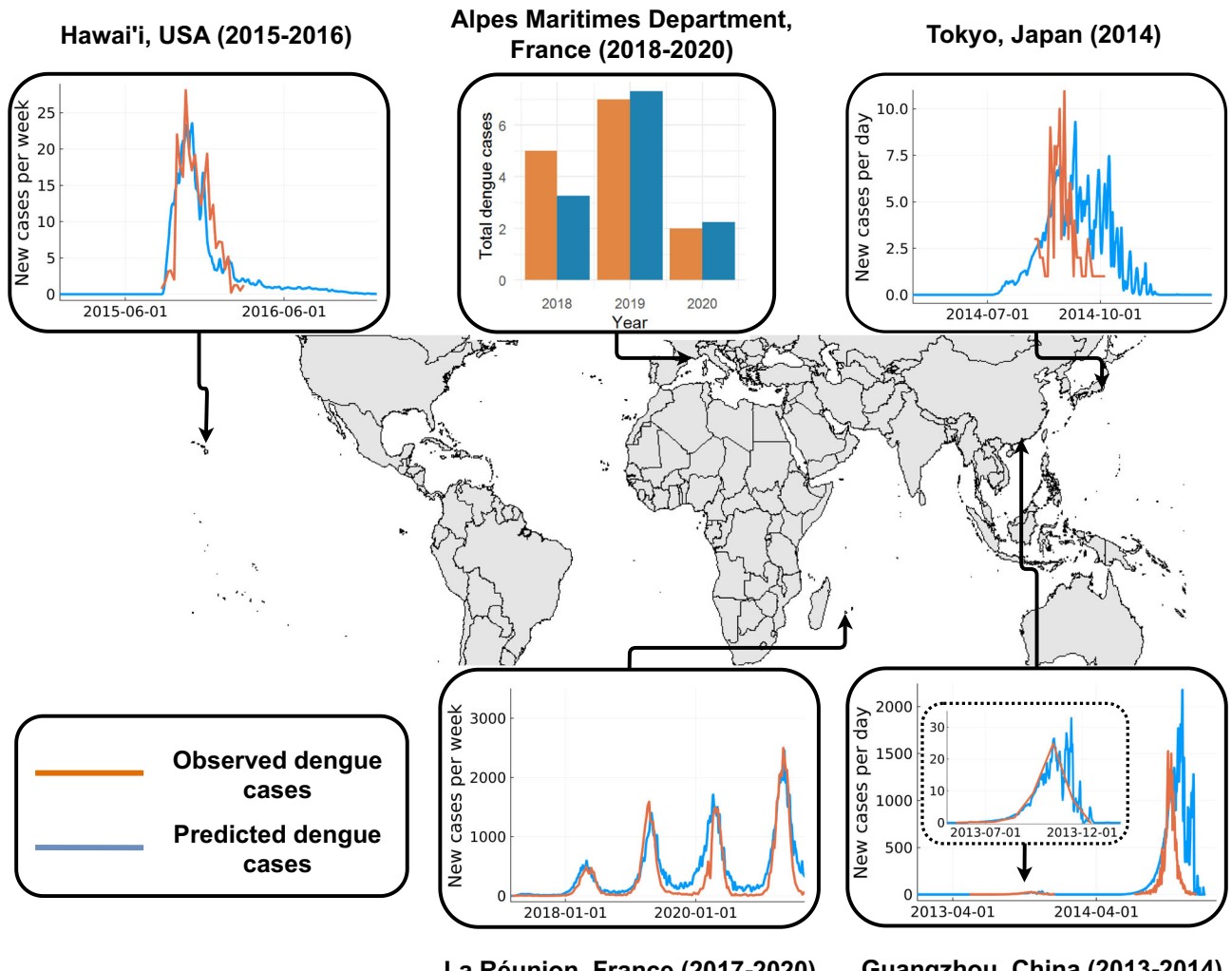

**Fig. 2 | Validation of predicted disease dynamics against historical outbreaks.** Comparisons of the number of instances of autochthonous transmission detected during historical dengue outbreaks and the model predictions[13,30,169–171]. For each outbreak the *x*-axis is time in days and the *y*-axis is the reported number of instances of autochthonous transmission, with the blue lines representing model predictions and the orange lines field observations from that same location. These results are discussed in detail in Supplementary Note 2 and source data are provided as a Source Data file.

Guangzhou in 2014, a considerable intensification of vector control activities occurred halfway through the outbreak to which, according to previous disease modelling studies, led to a substantial reduction in the final outbreak size and duration[28]. These validations demonstrate that, through the careful consideration of the mechanisms by which environmental variation acts on the life history of a vector species, broadly generalisable predictions of population and disease dynamics can be achieved.

**The role of phenotypic plasticity in driving disease outbreaks**
To examine how phenotypic plasticity contributes to the observed disease dynamics, we examine four of the outbreaks in Fig. 2 in greater detail. During each outbreak, we compare the wing lengths of all mosquitoes within the modelled population to the wing lengths of infected mosquitoes responsible for dengue virus transmission events (Fig. 3). Wing length is a trait that is observable in the field where it is known to seasonally vary in a manner that qualitatively matches with the variation we have observed in populations of *Ae. albopictus* and other mosquito species[17,34–36]. This variation is highly correlated with variation in adult longevity[9]. Adult longevity is used in the parametrisation of the model to link larval developmental experience to adult trait expression[9]. Large mosquitoes are produced as a result of low larval competition and cooler temperatures at the start of the adult season. These large mosquitoes have long lifespans which allows a greater proportion of individuals to survive through the extrinsic incubation period of the virus which is also prolonged when temperatures are low.

As each outbreak progresses the initial cohort of large mosquitoes begins to die off, being replaced by subsequent cohorts of smaller individuals that developed under higher intraspecific competition and warmer temperatures. Consequently, the wing length distribution of infected mosquitoes becomes increasingly similar to that of the entire population (Fig. 3A–D). This demonstrates that the trait dynamics of uninfected mosquitoes and mosquitoes responsible for infections are distinct, with the relationship between the two evolving through time and between outbreak locations. By considering differences between these wing length distributions we are able to directly examine the effects of phenotypic plasticity in adult longevity on disease dynamics[9]. For example, in Guangzhou during first quarter of the outbreak, an average of 70% of infections transmitted each day are attributable to the largest 20% of mosquitoes present in the population over this period (equivalently this is 47%, 31%, and 25% of transmission, attributable to the largest 20% of mosquitoes in Alpes-Maritimes, Reunion, and Hawai'i, respectively). More generally, across all of the outbreaks considered here we find that, in the initial stages of each outbreak, the majority of dengue virus transmission can be attributed

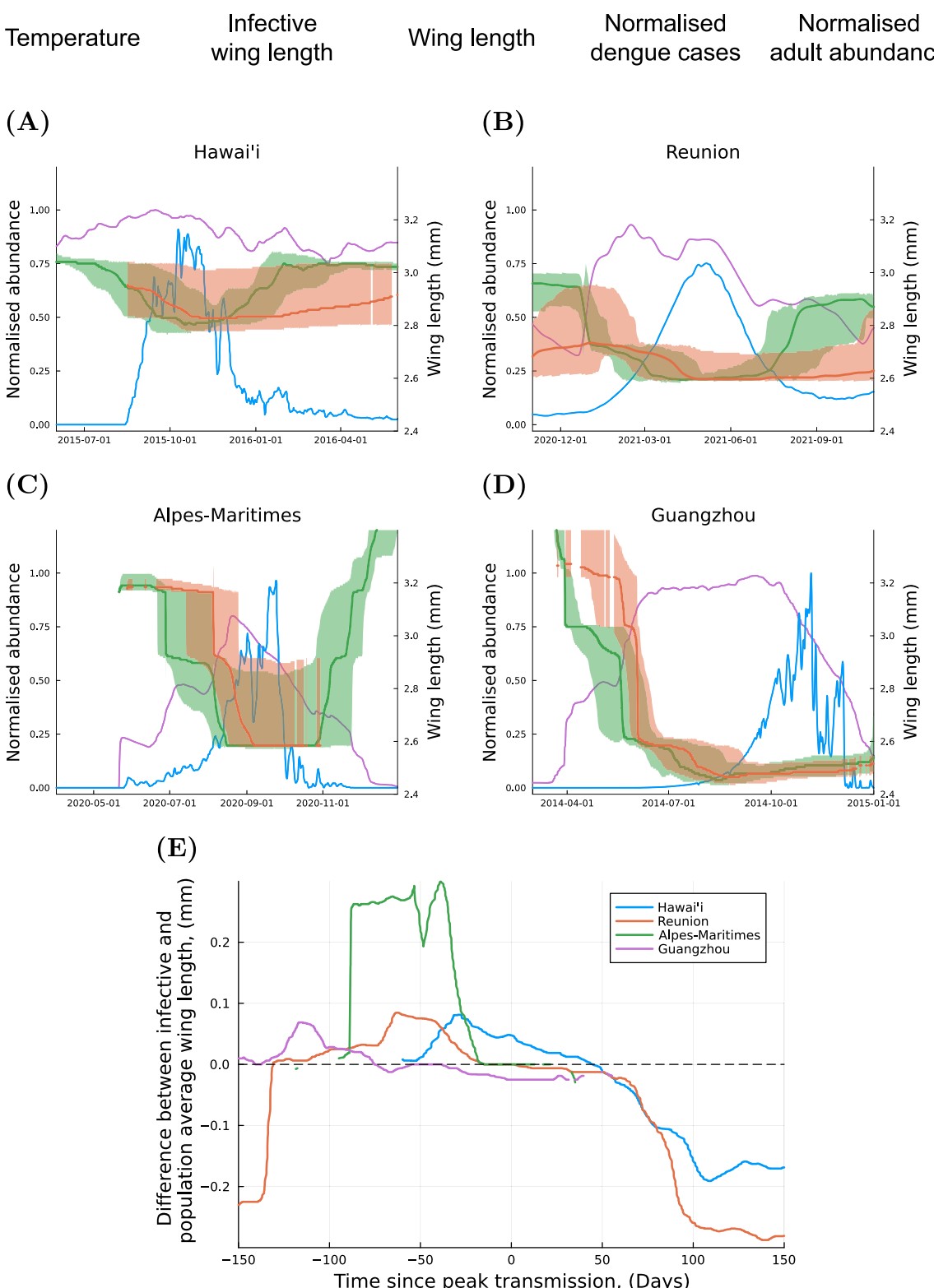

**Fig. 3 | Comparison of trait dynamics during historic dengue outbreaks.**
**A**–**D** Comparisons of the distribution of infective wing lengths and population average wing lengths during four historic dengue outbreaks. The central green line indicates the population's median wing length at a given time with the shaded area indicating the 25th and 75th percentiles of the wing length distribution. The central orange line indicates the 50th percentile of the distribution of wing lengths of infectious individuals with the shaded area indicating the 25th and 75th percentiles.

The blue and purple lines indicate the normalised number of dengue cases and adult mosquitoes respectively. Source data are provided as a Source Data file. Comparison for outbreaks in: **A** Hawai'i (2015–2016); **B** Réunion (2021); **C** Alpes-Maritimes, France (2020); **D** Guangzhou in (2014). **E** The difference between the average wing length of infective mosquitoes and the population average wing length in four locations before and after the time of peak transmission.

to mosquitoes with wing lengths that are higher than the population average (Fig. 3E).

In the transmission of dengue, extended longevity is the only advantage that we have assumed that large mosquitoes have over small mosquitoes. The disproportionate amount of disease transmission events caused by large individuals during the initial stage of each outbreak is therefore attributable to the effects of delayed environmental variation on the trait structure of vector populations. This result adds significant nuance to the seminal work of Macdonald[37] on malaria transmission, which demonstrated that under equilibrium conditions, adult longevity is an important factor in determining whether a VBD can spread. We show that temporal variation in adult longevity is essential for amplifying pathogen transmission in the early stages of an outbreak.

To test our hypothesis that vector trait dynamics are integral to predicting disease dynamics, we develop a suite of model variants (simplifications of our full model) that make common assumptions about how vector traits respond to environmental variation (see Supplementary Note 3). In the full model, we temporally track the number of mosquitoes within the population that express each wing length, resulting in a temporally evolving trait distribution, induced by delayed phenotypic plasticity. We consider a variant of our full model, where instead all adults are assumed to share the same invariant wing length, an assumption that is common in derivatives of the classical Ross-Macdonald transmission model, from which epidemiological metrics such as the basic reproduction number are routinely derived. We refer to this as the constant wing length model and consider its dynamics for different constant wing lengths[4]. Under this assumption, the only variation in adult traits occurs due to changes in the instantaneous environmental conditions, and so the effects of previous environments on the expression of traits are omitted. We also consider a model variant that uses the same delayed mechanisms of trait expression as the full model, but that represents this variation through a population average wing length rather than tracking the temporal evolution of the full trait distribution. We refer to this model variant as the unstructured model. Under this assumption, phenotypic plasticity is represented through a mean-field, where although the population's average past experience of development is accounted for, there is no explicit trait structure.

When the constant wing length model is simulated for the outbreak on Reunion (selected as a point of comparison due to its multi-year duration), under the assumption that all mosquitoes express the average wing length predicted by the full model (2.6 mm), we find that the constant wing length model predicts half as many total infections over the duration of the outbreak compared to the full model (see Supplementary Fig. 59). This demonstrates that the inclusion of mechanisms of individual variation not only allows us to attribute a disproportionate number of transmission events to large mosquitoes (Fig. 3), but also shows that the vector population's trait dynamics have a quantifiable effect on outbreak size, and therefore that phenotypic plasticity alters disease dynamics (see Supplementary Fig. 56 for examples of the constant wing length model for other outbreaks). For the same outbreak, the unstructured model variant predicts twice as many dengue cases as the full model. The mean-field approach, characterised by the unstructured model, does not account for the non-linear effect of adult survival on vector competence, resulting in quantitatively different predictions of outbreak size (Supplementary Fig. 59). These examples show how the delayed response of vector traits to past environments through phenotypic plasticity gives rise to a trait distribution that is integral to understanding the disease dynamics we observe. Further, they show how these disease dynamics cannot be anticipated by mean-field approaches, even when they incorporate all the same mechanisms of delayed phenotypic plasticity.

### Global risk

To predict transmission risk over the species global range, we derive an expression for the reproduction number, $R_t$, which describes the number of infections produced by the introduction of a single infected human into the population (described in the Methods and compared to the simpler model variants in Supplementary Note 3). This formulation of $R_t$ accounts for the effect of phenotypic plasticity in the vector population trait structure, on the ability of that population to transmit disease. It produces predictions that are generalisable between climates (see Methods for details). We compute $R_t$ across the species' global range and report the number of consecutive months per year between 2010–2020 that $R_t > 1$, in areas with a mean annual relative humidity greater than 55%, conditions that indicate the suitability of a region for the transmission of dengue virus and the survival of adult mosquitoes. In general, we predict less suitability for the autochthonous transmission of dengue vectored by Ae. albopictus than approaches that do not account for the delayed effects of phenotypic plasticity (See Supplementary Note 4 for a comparison between our predictions and those made by a standard formulation of the basic reproduction number $R_0$ that uses similar environmental drivers and reaction norms). Furthermore, by comparing the average period that $R_t > 1$ to the average density of adult female mosquitoes during the active season (Fig. 4A, C, E) we see that abundance alone does not predict the ability of vector populations to transmit disease. Additional outputs of parameters of interest, such as the time of peak adult density and the first and last days for which transmission is predicted can be found in Supplementary Note 5.

In North America and Europe, regions that currently experience limited autochthonous transmission of dengue virus by Ae. albopictus, our approach predicts that there is currently limited transmission risk from this vector. By comparison, in areas of China where dengue outbreaks vectored by Ae. albopictus are more frequent, our approach predicts longer transmission periods over a wider area, demonstrating that our predictions reflect currently observed regional differences in dengue incidence[38]. By comparing the locations of historical dengue outbreaks to locations that are predicted to be at risk, we observe that despite our predictions being highly specific, they still encompass locations known to experience autochthonous dengue virus transmission[33,39,40]. For example, the location of 38 of the 41 clusters of autochthonous dengue transmission in Europe reported by the European Centre for Disease Control since 2010 occurred within regions that we predict are at risk of dengue transmission (Fig. 4D)[13]. This suggests that over the last 10 years populations of Ae. albopictus in temperate regions have had limited capability to sustain dengue transmission due to environmental constraints despite the species now wide distribution in these regions. Given the changing nature of the climate, understanding when this widely distributed species will be capable of sustaining prolonged transmission cycles in temperate regions is of critical importance that is yet to be resolved, and which is the subject of subsequent modelling work.

## Discussion

In a world where the climate is changing rapidly, there is a clear need for accurate models for predicting outbreaks of environmentally sensitive, high-burden vector-borne diseases[41]. In a departure from the current state-of-the-art we show that stage-phenotypically structured delay differential equations are a practical and effective method for producing such predictions by performing the most comprehensive validation to date of a mechanistic model for the transmission of dengue virus by Ae. albopictus. We show that the delayed expression of phenotypic plasticity in response to previous environments plays a critical part in determining the ability of vector populations to transmit disease. This provides a rigorous theoretical basis to explore the long-theorised importance of phenotypic plasticity in determining disease

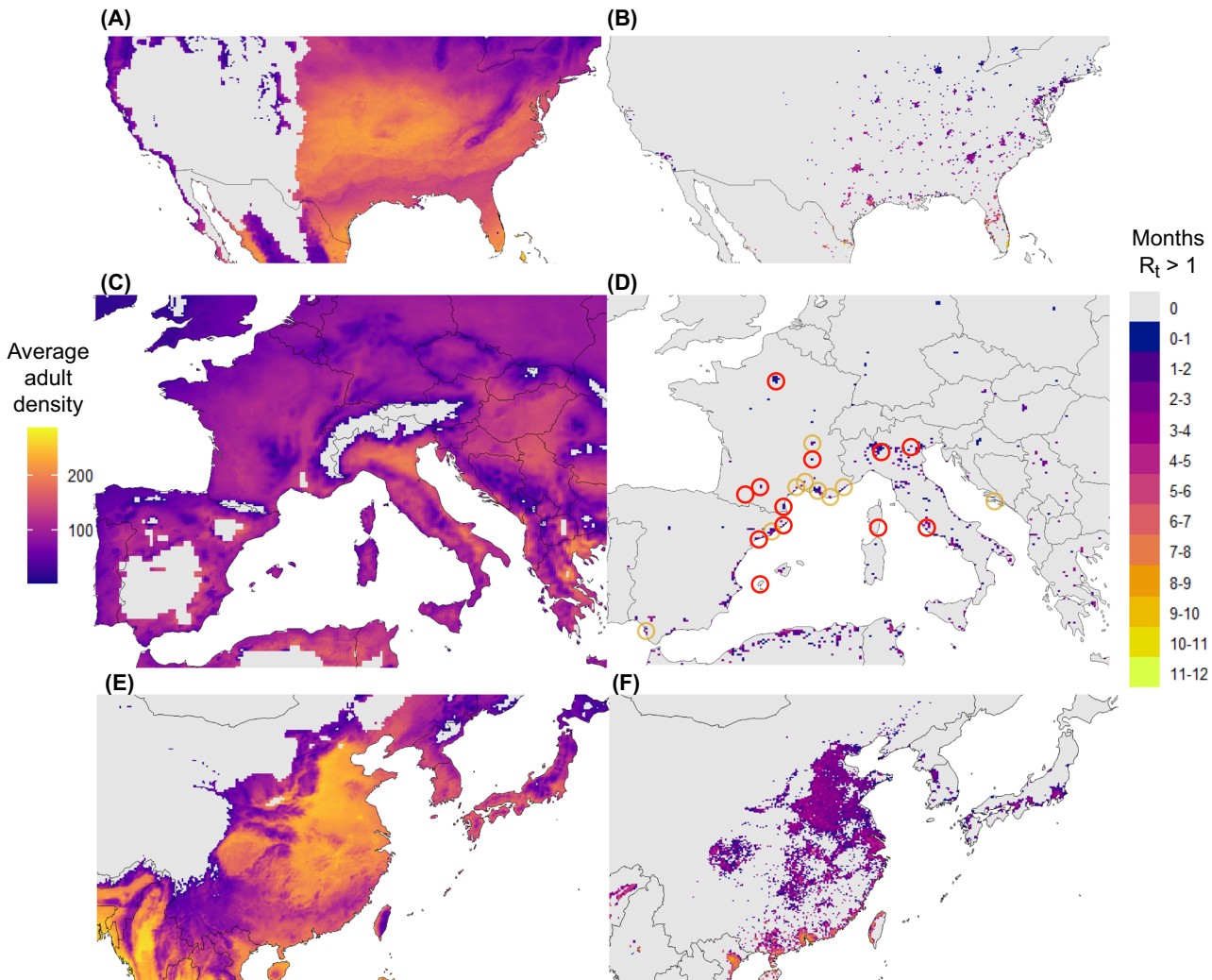

**Fig. 4 | Geographic variation in mosquito density and suitability for disease transmission. A, C, E** The average adult density during the active season of *Ae. albopictus* in North America, Europe, and Asia respectively between the years 2010−2019. **B, D, F** The average number of months the reproduction number was greater than one ($R_t > 1$) in North America, Europe, and Asia respectively between the years 2010−2019. In **D** the yellow circles are centred on regions within which autochthonous dengue transmission was detected between the years 2010 − 2019 and the red circles indicate the locations of outbreaks after this period. Source data are provided as a Source Data file.

risk that is widely applicable to other systems of vector-borne disease[4]. We find that vector trait variation plays a critical role in determining the ability of vector populations to sustain disease transmission and therefore that vector abundance alone is not sufficient to predict transmission risk. In light of our findings, vector traits may need to be better integrated into vector surveillance and research to facilitate more nuanced, robust predictions of population processes and transmission dynamics.

The ability of mechanistic modelling approaches to incorporate detailed vector ecology allows them to produce generalisable predictions of disease risk. However, incorporating these details requires substantial amounts of life-history data that does not yet exist for many vectors, favouring the use of less detailed mechanistic models than we develop here[42–44]. Even for well-studied species such as *Ae. albopictus*, there are data gaps in our current knowledge. Some of these gaps are life-stage specific, for example, when rearing adults for infection trials it is common to record the combined life-history traits of both larval and pupal life stages using a single generic class for juveniles, data which less detailed mechanistic approaches can readily utilise but that which we cannot. Although there are a sufficient number of studies exploring the effects of competition on larval life

history to produce well-parametrised larval reaction norms, there is very little available experimental data describing the life history of pupae which are rarely specifically reared for study. As each life stage may be affected differently to the same environmental stressor, life-stage-specific gaps in our knowledge of vector life history potentially mean that environmental limitations on these life stages are overlooked in our predictions of vector dynamics and consequently disease transmission dynamics. Similarly, although there has been a substantial effort to determine the lower thermal limits of the species' reaction norms due to the species' northward expansion, there is a more limited amount of life-history data available to describe the species' responses to the extremely high temperatures which are more frequent in the regions where the majority of dengue virus transmission occurs[14,45]. Although these limitations have not prevented our model from producing accurate predictions of population dynamics in the regions we consider, unless addressed they may limit the generalisability of our predictions of population dynamics to future climates, a factor that could be mitigated through modifications to standard experimental protocols.

When designing vector control campaigns to implement interventions such as the sterile insect technique, gene drive or transgenic

control, it is common to rely on mathematical models of vector population dynamics[46]. These models are used before implementing control measures to inform the required level of population suppression and the intensity of control measures necessary to achieve this target[47–49]. Similarly, after control measures have been implemented, mathematical models are a practical method to assess how effective an intervention has been by predicting how an outbreak would have progressed in the absence of control[50]. Without this additional analysis field-trials can only directly link the efficacy of their intervention to a reduction in mosquito abundance and not necessarily the reduction in human cases of disease that they aim to achieve, two factors we show here are not necessarily directly linked[51,52]. The 2014 outbreak of dengue in Guangzhou provides a well-studied example showing the potential of our approach as we and previous models are able to use mathematical models to assess the efficacy of the implemented control measures (see Fig. 2)[28,53]. However, for the majority of the outbreaks we considered, this sort of retrospective analysis is not possible due to a lack of detail in the reporting of the nature, timing, and intensity of control measures implemented. Further, to make reliable predictions to assist in the design of vector control it is imperative to use models that have been independently validated against real-world data from the target system[54]. Extensive and independent validation of mathematical models of systems of VBDs is currently not standard, despite the ready availability of surveillance data and is a critical step in ensuring these predictions are robust.

Dengue transmission frequently occurs in regions where *Ae. albopictus* co-occurs with the primary vector of dengue, *Ae. aegpyti*, and with which it competes for resources in larval habitats[55]. This complicates our ability to predict the effect of trait dynamics on transmission efficacy as the relationship between adult wing length, temperature, larval density, and food availability is asymmetrically altered by the degree of interspecific competition that individuals of either species experience[56–58]. Further, in endemic regions co-circulating serotypes and previous exposure to infection complicate transmission dynamics by introducing an immune structure to the human population[59]. An extension of our model to these areas is possible but would additionally require the development of a comparable model for *Ae. aegypti*, to quantify the effect that interspecific competition has on the life history of both species and then to extend the disease dynamics to consider multiple co-circulating serotypes of the dengue virus. However, even in these more complex transmission scenarios, the underlying interaction between environmental drivers and vector trait expression is driven by the same mechanisms of phenotypic plasticity that we explore here, as they are a fundamental feature of vector ecology. By developing our understanding of the drivers of disease dynamics that arise in regions with a single vector transmitting a single serotype of dengue virus, we are now well positioned to begin addressing the more complex dynamics underlying patterns of dengue transmission in other regions.

Current state-of-the-art methods for predicting VBD transmission often make implicit mean-field assumptions and assume that vector traits respond instantaneously to the current environment without consideration for the complex biological processes through which traits arise[60–67]. Trait-based approaches have been long advocated for as a way of producing generalisable predictions of population processes, but to date, there has been a limited application of theory to practice, in part due to insufficient life-history data and due to the increased complexity inherent in such approaches[68]. To produce robust predictions that reflect how complex systems adapt to climate change requires a fundamental shift in the way we account for the effects of individual variation. We must move away from mean-fields and towards a more complete representation of the mechanisms of phenotypic plasticity through which individuals vary and which drive population dynamics[10]. Deepening our understanding of the effects of individual variation on populations is critical beyond vector-borne

disease since mechanisms of individual variation are also important in determining the outcome of infectious disease dynamics, biological invasions, interspecific competition, and species responses to climate change[69–71]. The advantages of such an approach are clear and have allowed us to gain insights into a highly complex vector-borne disease system, shedding light on the role that phenotypic plasticity plays in shaping the ability of vector populations to spread disease.

## Methods

To predict both the trait and population dynamics of *Ae. albopictus* we apply the model framework developed in Brass et al.[12]. This framework combines a stage-structured population model of the form described in Nisbet and Gurney (1983) with an additional phenotypic structure that is able to represent the effects of within and between generation plasticity[72]. The phenotypic structure describes how an individual's previous experience of its environment during development alters the traits it currently expresses. Individuals with different experiences of the past environment are grouped into different environmental classes which describe how this experience alters the traits they express now. By linking individual-level variation to population response, we are able to predict not only the population's dynamics but also the population's trait structure. This approach is flexible and able to incorporate the effect, both instantaneous and delayed, of environmental stressors on multiple traits, making it ideal for accounting for the complex life history of *Ae. albopictus*.

The model predicts the dynamics of a population of mosquitoes arising from a single water body of fixed dimensions. The model inputs are environmental variables from the location being simulated and the outputs are predictions of population and trait dynamics (Fig. 5). Adult mosquitoes oviposit eggs either onto the surface of the water or around the sides of the habitat, with the proportion of eggs being placed around the side of the habitat increasing as water level decreases[73]. The eggs placed into or around the water body express either a diapausing or non-diapausing phenotype which is determined by a maternal effect in response to falling temperatures and decreasing photoperiod[24]. Egg diapause is a form of regular dormancy that allows eggs to withstand cold temperatures and is the primary mechanism by which *Ae. albopictus* populations persist and overwinter in temperate climates[74]. Once development is complete, both diapausing and non-diapausing eggs either become quiescent, a form of irregular dormancy that allows persistence through dry periods or immediately hatch into larvae[20]. Quiescence continues until the dormant egg is inundated by precipitation after which it immediately hatches.

In the model we assume that the water body only varies in response to changes in temperature, the accumulation of precipitation, and through evaporation of standing water and is otherwise identical in every respect between locations. We assume that all eggs initially either express the active or diapausing phenotype. All eggs expressing the active phenotype develop at a temperature-dependent rate after which they either hatch or become quiescent according to the water level in the habitat. After completing development, diapause eggs are assumed to remain dormant until a critical temperature and photoperiod is reached after which they immediately hatch or become quiescent[24]. We place restrictions on both the production of diapause eggs and upon their release from diapause to ensure that they are fully developed when they hatch. The survival of eggs through the egg stage is assumed to be determined solely by temperature with this relationship differing between egg phenotypes[75]. We assume all eggs in the quiescent egg class are identical regardless of previous phenotype and assume that quiescent eggs hatch when the water level rises.

Larval mosquitoes compete for available resources in the aquatic habitat and are assumed to develop at a rate determined by both the current temperature and the intensity of larval competition[9]. The ecology of larval mosquitoes is complex, and to produce a coherent model with the data currently available it is necessary to make broad

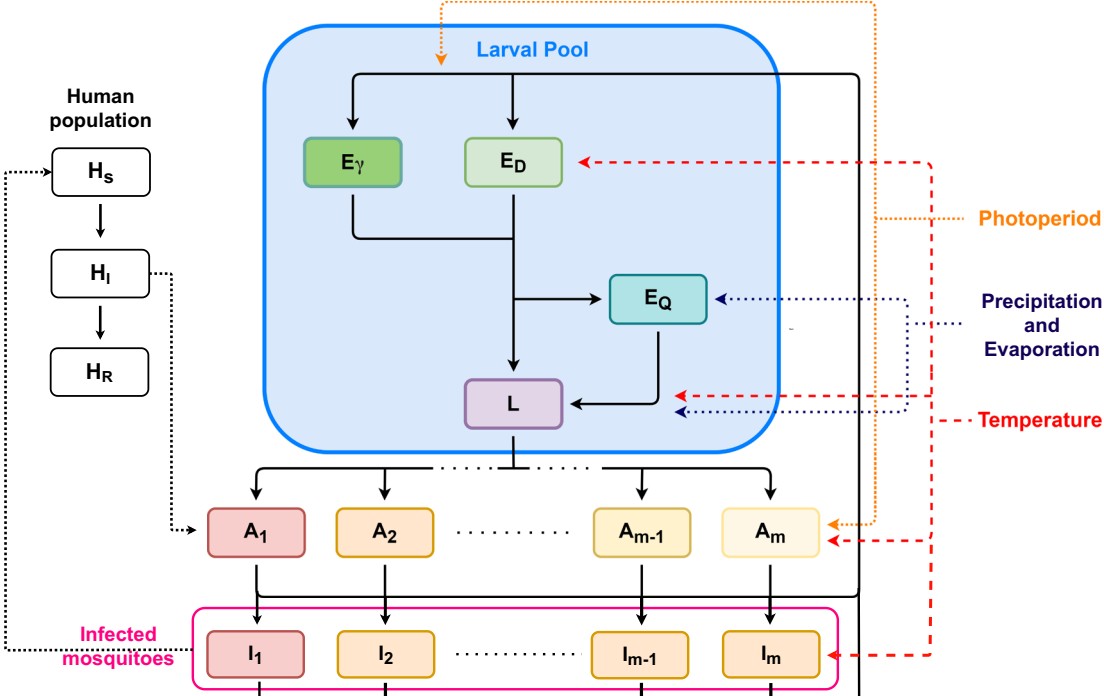

**Fig. 5 | Schematic of the model used to represent the transmission of dengue virus by *Ae. albopictus*.** We consider a stage-structured model with three explicitly modelled mosquito life stages, eggs, larvae, and adults, with pupae represented implicitly. The model considers the dynamics of a population from a single water body of a fixed dimension with a water level that is dependent solely on precipitation and evaporation. Eggs are divided into three different classes, active eggs $E_\gamma$, diapausing eggs $E_D$, and quiescent eggs $E_Q$, depending on the location they are placed in the pool (active/quiescent) and the phenotype they express (diapausing/non-diapausing). After hatching, eggs proceed to the larval class where development depends on temperature, intraspecific density, and the food available per larvae per day which is assumed to relate to the amount of water accumulated in the habitat and the current temperature. Once larval development is complete, larvae transition to the pupal class which is represented implicitly. This means, that although we account for the effect of environmental variation on the life history of pupae, this is accounted for without the need for state equations. Upon maturation to adulthood, each individual's experience of its environment as a larva determines the survival and fecundity that the individual expresses as an adult through developmental plasticity. To represent this wing length is discretised into $m$ environmental classes, with $m$ sufficiently large to avoid the effects of discretisation. Wing length is correlated with both adult longevity and fecundity and so is used to assign to each environmental class a set of adult traits, with adults in the $x^{th}$ environmental class denoted $A_x$. Upon infection by biting an infected human, $H_I$, uninfected mosquitoes in environmental class $j$, $A_j$, transition to the corresponding infected mosquito class, $I_j$. The infection cycle begins when a susceptible human, $H_S$, is bitten by an infected mosquito after which that human has a chance to become infected and move into the infected human class $H_I$. After the recovery period infected humans move into the resistant class, $H_R$, and cannot be infected again. The delay between exposure to infection and becoming infective is accounted for by the delay structure.

simplifying assumptions about the life history of larvae. We assume that all larvae are functionally identical, regardless of larval instar or previous experience of dormancy during the egg stage. Therefore, we only consider a single larval class[76]. It is assumed that there is no interspecific competition either for resources or through predation and that the only intraspecific competition is for resources and space[77,78]. Available resource is assumed to be consumed in its entirety and to be replenished daily, representing the product of temperature-dependent metabolic processes in the larval environment. Once larval development is complete pupation begins with the development of pupae assumed to depend solely on temperature[75]. The container habitats that *Ae. albopictus* prefer are generally small and vulnerable to flushing, a process whereby the body of water overflows and individuals are swept away[21]. We represent flushing by increasing the mortality of larvae and pupae whenever the height of the water in the habitat exceeds the height of the container and rainfall is sufficiently intense. These small containers are also susceptible to drying out, and whenever all water within the container evaporates, all non-quiescent juveniles are assumed to die out.

Adult mosquitoes experience developmental plasticity in response to their experience of temperature and intraspecific competition as larvae[79]. This is represented by tracking each individual's experience of the biotic and abiotic environment during development and using this to determine the traits that individuals express as adults.

Specifically, each individual's experience of the average temperature and the average food available per larvae per day over the course of the larval period is used to predict that individual's wing length. Wing length is then used to determine the fecundity and longevity of that individual as an adult[80]. Both fecundity and longevity are then further modified by the current temperature meaning that within the model adult traits respond to both current and historic environmental conditions[43,75]. The adult resources necessary to survive and complete the reproductive cycle, such as sugar and blood meals, are always assumed to be in excess and the model does not represent any effects of competition between adults[81]. The production of cold resistant diapausing eggs is triggered when falling temperatures and photoperiod reach a critical threshold[24].

### Model details
**The abiotic environment.** Abiotic environmental variation is incorporated into the model through temperature, photoperiod, evaporation, and precipitation from the location in which the modelled population is situated. Meteorological data is sourced from the ERA5-land climate reanalysis dataset[82]. Climatic variables in this dataset are available at a 0.1° × 0.1° resolution at an hourly time interval and are processed to produce a daily mean temperature and the total accumulated precipitation and evaporation each day. For use in the model each environmental variable is extended to continuous time using

splines, which were then tested against the climate data to ensure there were no errors in the interpolation. Splines were created using the function Spline1D from version 0.5. 2 of the Dierckx package, a wrapper from the Fortran library of the same name[83].

The temperature at time $t$ is denoted $T(t)$, and for simplicity, we assume that the temperature of the larval pool and the air temperature are always the same[84]. Temperature is used throughout the model to instantaneously alter the traits (survival and development rate) expressed by individuals in each developmental stage. Temperature plays a further role in two instances of delayed phenotypic plasticity represented in the model. The average temperature an individual experiences over the larval period in combination with the average amount of food available per larva per day determines the wing lengths of emerging adults. Also included is a maternal effect where the temperature and the photoperiod at time $t$, denoted $\psi(t)$, determines the proportion of produced eggs that express a diapausing phenotype. A combination of temperature and photoperiod subsequently determines when diapausing eggs are released from dormancy.

The total precipitation and evaporation each day, denoted $\rho(t)$ and $\epsilon(t)$ respectively, are used to simulate the dynamics of the aquatic developmental habitat. The model simulates the dynamics of a single cylindrical water body with a surface area of $\mu$ mm$^2$ and maximum volume of $V$ mm$^3$. Throughout the subsequent analysis these parameters are kept constant across the range of locations considered and have been selected to be consistent with productive habitats observed in the field[85]. Water accumulates in the habitat through precipitation and leaves through both evaporation and overspill. Overspill occurs when the amount of water in the pool exceeds the volume of the pool, at which point any additional precipitation is not added to the pools total volume. If an overspill event happens concurrently with an intense period of precipitation, a spike in mortality of the larval and pupal classes occurs, representing individuals being flushed away[21]. The total amount of water in the habitat on day $t$ after all precipitation and evaporation can therefore be expressed by the time series

$$W_t = \min\{W_{t-1} + \mu(\rho(t) - \epsilon(t)), V\}, \qquad (1)$$

with an initial condition such that the habitat is full at the start of the simulation, i.e., $W_0 = V$. This time series for daily water level is then extended to continuous time using a spline, such that the water level at time $t$ is denoted $W(t)$. Whenever $W(t) = 0$ the container has dried out and all non-quiescent juvenile stages (active eggs, diapausing eggs, larvae, and pupae) die-off. Habitat water volume, $W(t)$, also governs the proportion of eggs produced that are quiescent and when they are released from dormancy.

**Eggs.** The number of active, quiescent and diapausing eggs present in the population at time $t$ is denoted by $E_\gamma(t)$, $E_Q(t)$ and $E_D(t)$ respectively. Active eggs develop at a temperature-dependent rate, immediately hatching once development is complete if they are at or below the current water level and moving into the quiescent class otherwise[75]. Quiescent eggs remain dormant until inundated by precipitation after which they immediately hatch[20]. Hatched eggs enter the larval class. Diapausing eggs are cold resistant and are produced in response to a maternal effect. They remain dormant until a critical temperature and photoperiod threshold is reached after which they either immediately hatch or become quiescent[24]. In other *Aedes* species, it has been demonstrated that adult body size is related to egg size and that larger egg sizes increase desiccation resistance[86,87]. Although weak effects of egg size on survival have been observed in *Ae. albopictus*, there is insufficient empirical data to justify the inclusion of such a function in this model. Therefore, it is assumed that the only maternal effect is whether an egg is active or diapausing[88].

For active eggs, our model includes two state-equations that describe how the stage duration and through-stage survival of active

eggs vary through time, which we denote by $\tau_{E_\gamma}(t)$ and $S_{E_\gamma}(t)$, respectively. To describe how these quantities evolve we parametrise two temperature-dependent reaction norms $g_{E_\gamma}(T)$ and $\hat{S}_{E_\gamma}(T)$ which describe the development rate of active eggs and the proportion of active eggs that survive through the stage when held at constant temperature $T$. Note that $\hat{S}_{E_\gamma}(T)$ is distinct from $S_{E_\gamma}(t)$, the former is a function of temperature describing the reaction norm, the latter is a function of time, describing time-dependent survival. We use a similar notation for survival and reaction norms of the other life stages. From the underlying Nisbet and Gurney (1983) framework we inherit a relationship between the development rate, the probability of through-stage survival and the mortality rate. We therefore use the temperature-dependent reaction norms for development and through-stage survival to define an expression for the temperature-dependent mortality rate of active eggs, $\delta_{E_\gamma}(T) = -\log(\hat{S}_{E_\gamma}(T))g_{E_\gamma}(T)$[72]. The forms of the reaction norms used for egg survival and development were selected by fitting function forms that have been used in previous models to life-history data. We trial quadratic and Briere functional forms for the development rates, and quadratic and Gaussian functional forms for through-stage survival with each functional form being truncated at a value close to 0[8]. Functional forms were fitted using a weighted non-linear least squares approach and AIC was used for model selection, using the nls function in R[89]. The parameter values are reported in Supplementary Note 6, and the data used to perform this parametrisation was taken from a variety of published constant temperature laboratory experiments[75,90–97]. The best fitting functional form for the development rate of active eggs was a quadratic given by

$$g_{E_\gamma}(T) = \max\{\sigma_{11}T^2 + \sigma_{12}T + \sigma_{13}, 0.01\}. \qquad (2)$$

The best model for the through-stage survival of active eggs was a Gaussian with a functional form

$$\hat{S}_{E_\gamma}(T) = \frac{\sigma_{21}}{\sigma_{22}\sqrt{2\pi}}\exp\left(-\frac{1}{2}\left(\frac{T-\sigma_{23}}{\sigma_{22}}\right)^2\right). \qquad (3)$$

The result of this fitting is provided in Fig. 6A and B respectively.

Egg quiescence is a mechanism of irregular dormancy that allows a reservoir of eggs to persist through periods when the habitat has completely dried out[20]. After maturing out of the active stage, a proportion of eggs are assumed to enter the quiescent egg class dependent upon the water level within the container habitat. Quiescent eggs contain fully developed pharate larvae that remain dormant inside the egg until inundated by water, and therefore within the quiescent egg class, no development occurs. The proportion of eggs that enter the quiescent class at time $t$ is represented by the function $Q(t)$ which is defined by

$$Q(t) = \begin{cases} \frac{V-W(t)}{V}, & \text{if } W(t) < W(t-1) \\ 0, & \text{otherwise}. \end{cases} \qquad (4)$$

This function links the proportion of eggs that become quiescent to the height of the water in the container habitat. The condition $W(t) < W(t-1)$ ensures eggs only become quiescent if the water level on day $t$ is lower than that on day $t-1$. This choice of functional form ensures that when the habitat completely dries out, $W(t) = 0$, all eggs become quiescent upon maturation and so $Q(t) = 1$. As the water level rises a lower proportion of eggs enter the quiescent class until the habitat is completely filled, $W(t) = V$, and therefore $Q(t) = 0$. Within the quiescent egg class, it is assumed that the same temperature-dependent mortality rate used for active eggs is suitable for quiescent eggs.

As egg quiescence is an irregular form of dormancy, maturation out of the quiescent class is triggered by inundation rather than the

(A)

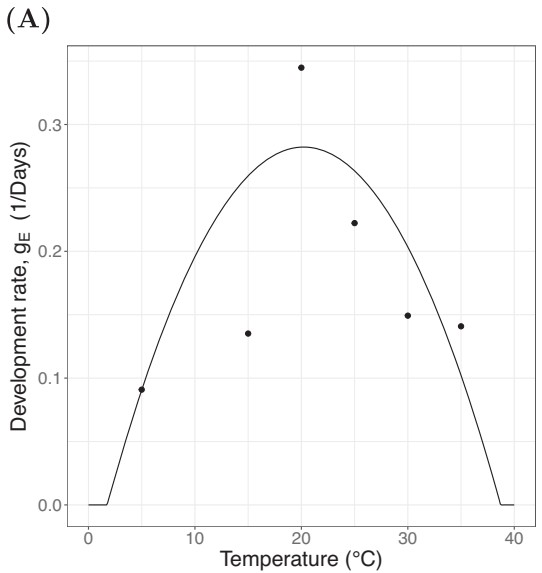

(B)

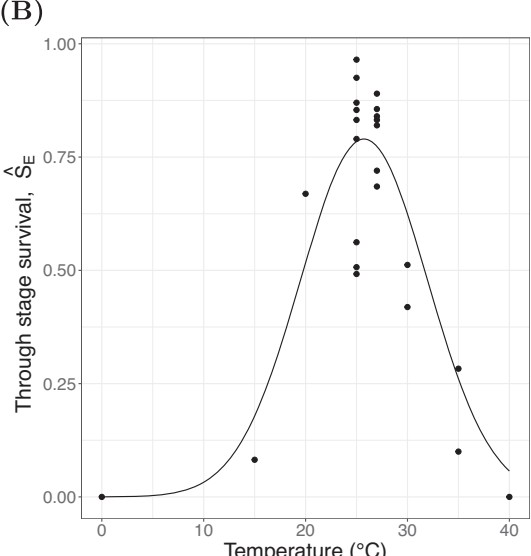

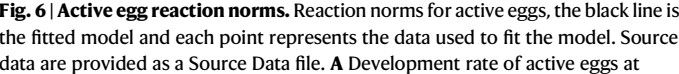

**Fig. 6 | Active egg reaction norms.** Reaction norms for active eggs, the black line is the fitted model and each point represents the data used to fit the model. Source data are provided as a Source Data file. **A** Development rate of active eggs at temperature $T$, $g_{E_\gamma}(T)$. **B** Through egg-stage survival of active eggs held at constant temperature $T$, $\hat{S}_{E_\gamma}(T)$.

completion of development. This is represented through the function $h_Q(t)$, which describes the rate at which eggs that hatch out of the quiescent class at time $t$ and is defined by

$$h_Q(t) = \begin{cases} 1 \times \left(1 - \frac{V - W(t)}{V}\right), & \text{if } W(t) > W(t-1) \\ 0, & \text{otherwise}. \end{cases} \quad (5)$$

When the habitat is completely full, $W(t) = V$, and so $h_Q(t) = 1$ and all quiescent eggs are inundated and hatch over the course of a day. The condition $W(t) > W(t-1)$ ensures that no eggs are released from quiescence when the water level at time $t$ is lower than the water level at time $t-1$ and therefore no new eggs would become inundated.

Egg diapause is a mechanism of regular dormancy that allows the persistence of populations through cold winter months and is governed by a maternal effect[24]. This is represented in this model by the diapause function, $D(t)$, which describes the proportion of adults at time $t$ that are producing non-diapause eggs and is defined by

$$D(t) = \begin{cases} 1 - \frac{1}{1 + 15e^{\psi(t,l) - \phi(l)}}, & \text{if } T(t) \leq 18, \text{ and } \psi(t) \leq \psi(t-1) \\ 1, & \text{otherwise} \end{cases} \quad (6)$$

with the functional form taken from Lacour et al.[24] and where $\phi(l)$ is the critical photoperiod threshold required to induce the production of diapausing eggs in adults at latitude $l$ defined by

$$\phi(l) = 0.1|l| + 9.5 \quad (7)$$

according to the relationship determined by Armbruster et al.[74]. The emergence of eggs from diapause is triggered by the current environmental conditions exceeding a critical temperature and photoperiod threshold[98]. We assume that once the environmental thresholds are reached, all eggs are released from diapause hatch over the course of a day. This is represented by the function $h_D(t)$ which is defined by

$$h_D(t) = \begin{cases} 1, & \text{if } T(t) \geq 12.5, \text{ and } \psi(t) > \phi(l), \text{ and } \psi(t) > \psi(t-1) \\ 0, & \text{otherwise}. \end{cases} \quad (8)$$

The mortality rate of diapausing eggs is defined by

$$\delta_{E_D}(t) = \begin{cases} 0.01, & \text{if } T(t) > -12 \\ 0.1, & \text{otherwise}. \end{cases} \quad (9)$$

This choice of mortality rate allows diapausing eggs to survive exposure to extreme temperatures for short periods[45]. This ensures that brief periods of intense cold do not kill off the whole population whilst ensuring that sustained low-temperature conditions remain unsuitable.

**Larvae.** Larval *Ae. albopictus* consumes a range of bacteria, plants and detritus. Both the time that larvae take to develop and the proportion of individuals that survive through the larval stage is assumed to depend on the intensity of intraspecific competition for resource in addition to temperature[9]. Predicting how much resource is available for larval *Ae. albopictus* in a container habitat from climate data alone would be a significant undertaking for even a limited geographical range. Resource dynamics are complex, and the amount of resources available in a container habitat varies with location, micro-climate, and community composition amongst many other factors that are not accounted for here[99]. Instead, we assume that the amount of food available in the container is temperature-dependent, proportional to the volume of water in the habitat and completely independent of location. The gross primary productivity of the larval habitat is then defined to be

$$F(t) = 10^{-6} \log_{10}(0.45 + 0.095T(t))W(t) + f_d \quad (10)$$

where the logarithmic term represents the products of respiration following the work of White et al. and the constant term, $f_d$, is a reservoir of nutrition from the decay of detritus[100]. It is further assumed that larvae divide the available food equally and completely (i.e., scramble competition)[101]. The amount of food available per larvae per day, $\alpha(t)$, can therefore be expressed by

$$\alpha(t) = \frac{F(t)}{L(t)}. \quad (11)$$

(A)

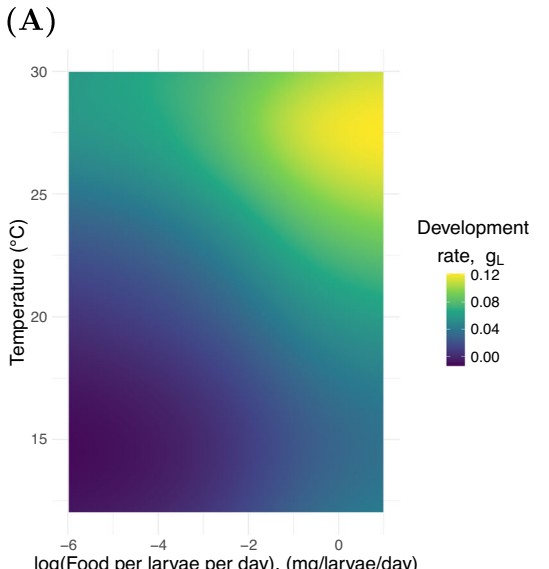

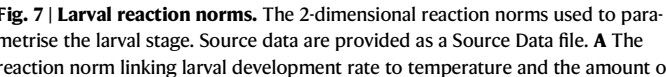

(B)

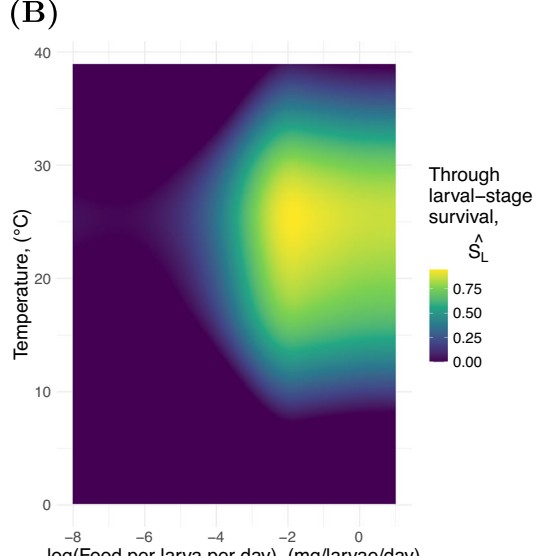

**Fig. 7 | Larval reaction norms.** The 2-dimensional reaction norms used to parametrise the larval stage. Source data are provided as a Source Data file. **A** The reaction norm linking larval development rate to temperature and the amount of food available per larva per day. **B** The reaction norm linking through larval stage survival to temperature and the amount of food available per larva per day.

As with active eggs, we include two state equations that describe how the stage-duration and through-stage survival of larvae changes through time which we denote, $\tau_L(t)$ and $S_L(t)$, respectively. The functions used to represent larval life-history processes are comprised of multiple components but are principally constructed from reaction norms for larval development rate, $g_L(T, \alpha)$, and through-stage survival independently of hydrological processes for larvae held at constant temperatures and resource densities, $\hat{S}_L(T,\alpha)$. To fit the multi-dimensional reaction norms for larvae we use generalised additive mixed effect models (GAMMs) using the R package gamm4 version 0.2. 6[102]. Both the life-history data used to parametrise these reaction norms and the code to fit the models can be found in the repository[89,102]. We consider models that include different combinations of temperature, density, and an interaction term between the two variables. As the data used to parametrise these functions are sourced from laboratory experiments that used populations of *Ae. albopictus* from different origin locations and reared larvae on different diets, these factors are included as random effects. For each life-history trait, the model with the lowest AIC is selected. The results for larval development rate and through-stage survival are shown in Fig. 7. The data used for this fitting was compiled from published laboratory experiments with both constant temperature and constant food provided per larvae per day[75,79,90,93–97,103–118].

The larval mortality term, $\delta_L(T(t), \alpha(t))$, accounts for the dynamics of the container habitat and density dependence. The water level of the container habitat varies with evaporation and precipitation. Spikes in larval mortality can therefore either be induced through the habitat drying out or through the flushing of larvae when the habitat overflows. This is represented in the larval mortality rate by setting $\delta_L(t) = \delta_d$ whenever the habitat dries out, (i. e., $W(t) = 0$) and similarly setting $\delta_L(t) = \delta_f$ whenever the habitat overflows and the intensity of rainfall is sufficient to initiate flushing (i. e., $W(t) = V$ and $\mu\rho(t) > 0.5 V$). The final component of the larval mortality function is an overcrowding term that increases the mortality of larvae when the larval density exceeds 3 larvae per ml according to a Gompertz function[119]. High larval densities are unfavourable for survival, and the term is necessary in addition to the reaction norms to ensure that the fluctuating water level does not cause implausibly high larval densities[120].

Therefore, the expression for the mortality rate of larvae is defined by

$$
\delta_L(t) = \begin{cases} \min\left\{-\log(\hat{S}_L(T(t),\alpha(t)))g_L(T(t),\alpha(t)) + \exp\left(-\exp\left(\frac{1-L(t)/3}{W(t)/1000}\right)\right),1\right\}, \\ \quad \text{if} \quad 0 < W(t) \le V \quad \text{and} \quad \mu\rho(t) \le 0.5V, \\ \delta_d, \quad \text{if} \quad W(t) = 0, \\ \delta_f, \quad \text{if} \quad W(t) = V \quad \text{and} \quad \mu\rho(t) > 0.5V. \end{cases}
$$

(12)

**Pupae.** The pupal development rate and survival are assumed to depend solely on temperature as there is a lack of experimental data to quantify the effect of developmental plasticity from the larval stage. We include two state equations that describe how the stage duration and through-stage survival of pupae vary, denoted $\tau_P(t)$ and $S_P(t)$ respectively. To describe how these quantities vary we once again parametrise reaction norms for the development rate and survival of pupae held at constant temperature $T$, denoted by $g_P(T)$ and $\hat{S}_P(T)$ respectively with the forms of the reaction norms shown in Fig. 8. These reaction norms are fitted using the same procedure described for the egg stage, with a Briere function best describing the relationship between pupal development rate and temperature with form

$$
g_P(T) = \max\left\{\sigma_{31}T(T - \sigma_{32})(\sigma_{33} - T)^{1/\sigma_{34}}, 0.01\right\}.
$$

(13)

A quadratic functional form is chosen for the relationship between through-pupal stage survival and temperature independently of hydrological processes

$$
\hat{S}_P(T) = \max\left\{\sigma_{41}T^2 + \sigma_{42}T + \sigma_{43}, 0.01\right\},
$$

(14)

parametrised using data from various sources[75,93,95–97,106,109,121,122].

The survival and mortality parameters are modified as in the larval stage to induce mortality when $W(t) = 0$ and to account for flushing such that

$$
\delta_P(t) = \begin{cases} \frac{-\log(\hat{S}_P(T(t)))}{\tau_P(T(t))}, & \text{if} \quad 0 < W(t) \le V \quad \text{and} \quad \mu\rho(t) \le 0.5V, \\ \delta_d, & \text{if} \quad W(t) = 0, \\ \delta_f, & \text{if} \quad W(t) = V \quad \text{and} \quad \mu\rho(t) > 0.5V. \end{cases}
$$

(15)

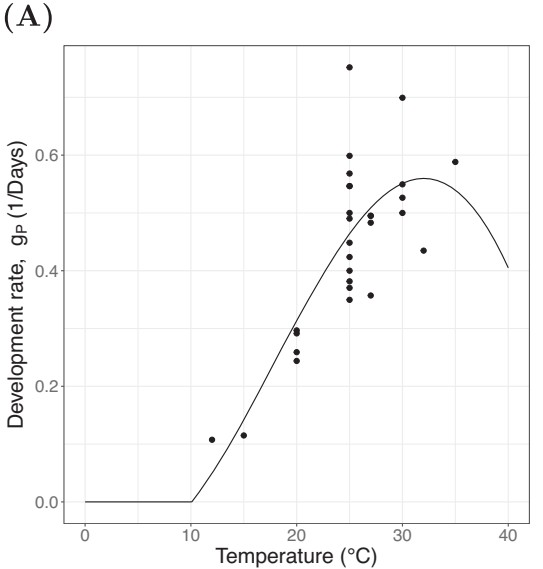

**Fig. 8 | Pupal reaction norms.** The reaction norms used to parametrise the pupal stage, the black line is the fitted model and the points represent the data used to fit the model. Source data are provided as a Source Data file. **A** The reaction norm linking the duration of the pupal stage to temperature. **B** The reaction norm through-pupal stage survival.

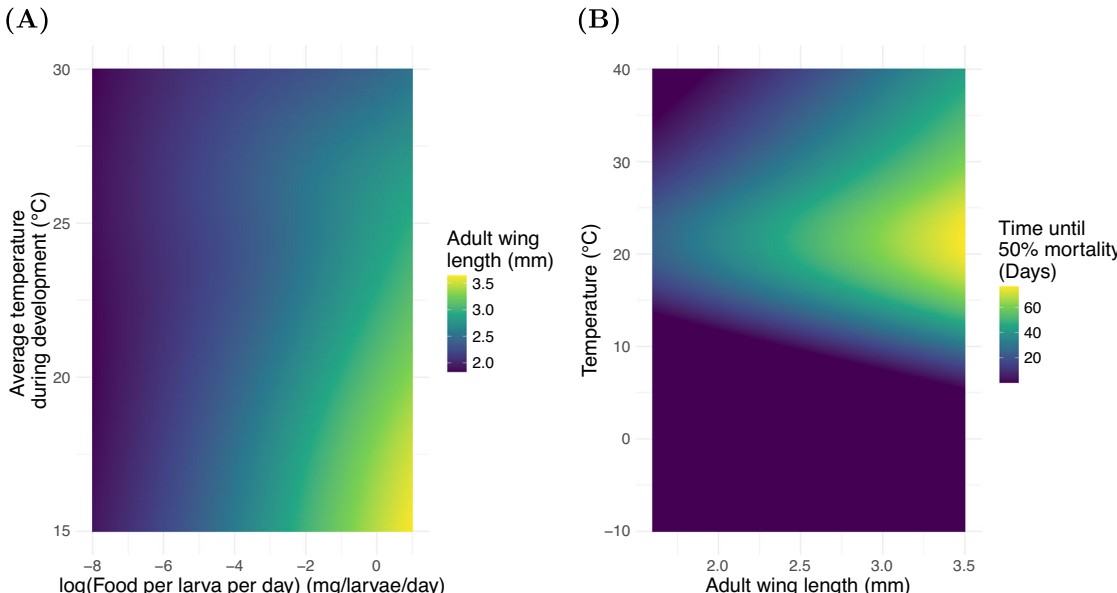

**Fig. 9 | Adult reaction norms.** The reaction norms used to parametrise the adult stage. Source data are provided as a Source Data file. **A** The reaction surface linking historical experience of larval competition for food, $\bar{\alpha}$, and temperature, $T_{avg}$, to adult wing length. **B**, The relationship between adult wing length the current temperature and adult longevity. This is converted into a mortality rate for use in the model.

**Adults.** *Ae. albopictus* experiences developmental plasticity in response to the temperature, food availability, and conspecific density experienced during juvenile development that contributes towards temperature-dependent mortality and fecundity of adults[9,80]. This is included in the model by using an individual's experience of larval competition and temperature through development to predict the wing length that the individual will express as an adult. This wing length is then used along with the current temperature to define the mortality rate of adults and the rate at which eggs are produced. This overlooks a known crowding effect whereby high densities of larvae inhibit development independently of competition for resource[79]. This choice is necessary as much of the experimental data used to parametrise reaction norms does not control for larval density and food availability simultaneously and therefore this relationship cannot be included in the model. Also omitted are the effects of the availability of adult resources on the production of eggs, and other varied processes demonstrably important in the life history of adults[118,123]. Using the same GAMM fitting procedure as described for larvae, we fit a 2-dimensional reaction norm for the wing length of emerging adults, $w_L(T_{avg}(t), \bar{\alpha}(t))$, as predicted by the average larval temperature and food available per larvae per day over the course of the larval period (Fig. 9A). We also fit the reaction norm for the time until 50% adult mortality, $\tau_{A_{50}}(w_L, T(t))$, as predicted by wing length and temperature (Fig. 9B). The parameterisations are carried out using data from

published laboratory data from experiments that used constant temperatures[5,9,79,90–94,96,108–111,113,115–118,121–143].

In the model, the effects of developmental plasticity are represented by assigning each individual to an environmental class that reflects their experience of the environment through development. This is achieved by partitioning the adults into $m$ environmental classes that define the wing length, $w_L$, the adults are predicted to express based upon their experience of temperature and resource availability as larvae. We divide the possible wing lengths an adult could express, $(w_{min}, w_{max})$ into $m$ equally spaced subintervals and denote by $w_j$ the midpoint of the $j^{th}$ sub-interval. Then define the function $g$ such that $g(w) = w_j$ if $w \in \mathbb{R}$ is in the $j^{th}$ subinterval of $(w_{min}, w_{max})$.

To determine the average amount of food available for larvae maturing into adulthood at time $t$, we define

$$\bar{\alpha}(t) = \frac{\int_{t-\tau_P(t)-\tau_L(t-\tau_P(t))}^{t-\tau_P(t)} \frac{F(s)}{L(s)} ds}{\tau_L(t-\tau_P(t))}. \qquad (16)$$

Then, we define the transition function, $w_j(T_{avg}(t), \bar{\alpha}(t))$, which determines which adult class a fully developed juvenile is assigned to, with

$$w_j(T_{avg}(t), \bar{\alpha}(t)) = \begin{cases} 1, & \text{if } g(w(T_{avg}(t), \bar{\alpha})(t)) = w_j \\ 0, & \text{otherwise} \end{cases}$$

for $j \in 1, \ldots, m$.

The fecundity of an individual expressing wing length $w$ is then given by

$$q(w) = 0.5 \exp^{\sigma_{51} + \sigma_{52} w}, \qquad (17)$$

noting that a factor of 0.5 is required as only females produce eggs[80]. The duration of a gonotrophic cycle is assumed to be temperature-dependent and to be the reciprocal of the biting rate. This is parameterised using a Briere function taken from Mordecai et al.[8] of the form

$$G(T(t)) = \frac{1}{\max\left\{\sigma_{61} T(T - \sigma_{62})(\sigma_{63} - T)^{\frac{1}{2}}, 0.01\right\}}, \qquad (18)$$

from which the rate at which an individual with wing length $w$ produces eggs at temperature $T(t)$ can be determined. The number of eggs an adult in environmental class $j$ produce eggs over the course of a gonotrophic cycle is defined by

$$q_j = q(w_j). \qquad (19)$$

Similarly, for individuals in environmental class $j$ that express wing length $w_j$ the mortality rate of adults is defined to be

$$\delta_{A_j}(T(t)) = \frac{-\log(0.5)}{\tau_{A_{50}}(w_j, T(t))}. \qquad (20)$$

**Population model.** Following the model framework defined in Brass et al.[12], based upon the work of Nisbet and Gurney[72], the stage-structured model to predict the population dynamics of *Ae. albopictus* is

$$\frac{dE_\gamma(t)}{dt} = R_{E_\gamma}(t) - M_{E_\gamma}(t) - \delta_{E_\gamma}(t) E_\gamma(t), \qquad (21)$$

$$\frac{dE_D(t)}{dt} = R_{E_D}(t) - M_{E_D}(t) - \delta_{E_D}(t) E_D(t), \qquad (22)$$

$$\frac{dE_Q(t)}{dt} = R_{E_Q}(t) - M_{E_Q}(t) - \delta_{E_Q}(t) E_Q(t), \qquad (23)$$

$$\frac{dL(t)}{dt} = R_L(t) - M_L(t) - \delta_L(t) L(t), \qquad (24)$$

$$\frac{dA_j(t)}{dt} = R_{A_j}(t) - \delta_{A_j}(t) A_j(t), \quad \text{for } j \in 1, \ldots, m. \qquad (25)$$

The variable developmental delay terms are defined such that

$$\frac{d\tau_{E_\gamma}(t)}{dt} = 1 - \frac{g_{E_\gamma}(t)}{g_{E_\gamma}(t - \tau_{E_\gamma}(t))}, \qquad (26)$$

$$\frac{d\tau_L(t)}{dt} = 1 - \frac{g_L(t)}{g_L(t - \tau_L(t))}, \qquad (27)$$

$$\frac{d\tau_P(t)}{dt} = 1 - \frac{g_P(t)}{g_P(t - \tau_P(t))}. \qquad (28)$$

The through-stage survival terms are defined such that

$$\frac{dS_{E_\gamma}(t)}{dt} = S_{E_\gamma}(t) \left( \frac{g_{E_\gamma}(t) \delta_{E_\gamma}(t - \tau_{E_\gamma}(t))}{g_{E_\gamma}(t - \tau_{E_\gamma}(t))} - \delta_{E_\gamma}(t) \right), \qquad (29)$$

$$\frac{dS_L(t)}{dt} = S_L(t) \left( \frac{g_L(t) \delta_L(t - \tau_L(t))}{g_L(t - \tau_L(t))} - \delta_L(t) \right), \qquad (30)$$

$$\frac{dS_P(t)}{dt} = S_P(t) \left( \frac{g_P(t) \delta_J(t - \tau_P(t))}{g_P(t - \tau_P(t))} - \delta_P(t) \right). \qquad (31)$$

Recruitment and maturation terms defined by

$$R_{E_\gamma}(t) = D(t) \left( \frac{\sum_{j=1}^m q_j A_j(t)}{G(t)} \right) + C_E(t), \qquad (32)$$

$$M_{E_\gamma}(t) = \frac{g_{E_\gamma}(t)}{g_{E_\gamma}(t - \tau_{E_\gamma}(t))} R_{E_\gamma}(t - \tau_{E_\gamma}(t)) S_{E_\gamma}(t), \qquad (33)$$

$$R_{E_D}(t) = (1 - D(t)) \left( \frac{\sum_{j=1}^m q_j A_j(t)}{G(t)} \right), \qquad (34)$$

$$M_{E_D}(t) = h_D(t) E_D(t), \qquad (35)$$

$$R_{E_Q}(t) = Q(t)(M_{E_\gamma}(t) + M_{E_D}(t)), \qquad (36)$$

$$M_{E_Q}(t) = h_Q(t) E_Q(t), \qquad (37)$$

$$R_L(t) = (1 - Q(t))(M_{E_\gamma}(t) + M_{E_D}(t)) + M_{E_Q}(t), \qquad (38)$$

$$M_L(t) = \frac{g_L(t)}{g_L(t - \tau_L(t))} R_L(t - \tau_L(t)) S_L(t), \qquad (39)$$

$$R_{A_j}(t) = w_j(T_{avg}(t), \bar{\alpha}(t)) \frac{g_P(t)}{g_P(t - \tau_P(t))} M_L(t - \tau_P(t)) S_P(t), \text{ for } j \in 1, \dots, m, \tag{40}$$

where $C_E(t)$ is an impulse function that introduces active eggs to the system at time 0.

The initial conditions are $E_D(0) = 100$, $E_\gamma(0) = E_Q(0) = L(0) = A_j(0) = 0$ for $j \in 1, \dots, m$, $S_X(0) = \exp\{-\tau_X(T(0))\delta_X(T(0))\}$ for $X \in E, L, P$. The initial history is for all $t \le 0$ and all simulations are performed with $m = 200$ unless otherwise noted. The model was simulated in Julia version 1.8 using the package DifferentialEquations[144,145].

## SEIR Model

The model for the population and trait dynamics of *Ae. albopictus* is incorporated into a susceptible-exposed-infected-resistant (SEIR) model for dengue virus vectored by *Ae. albopictus* (Fig. 5). We specifically consider the introduction of a single serotype of dengue virus into a completely susceptible human population. This simplified representation of the urban dengue transmission cycle is most applicable to non-endemic regions, such as Europe, the United States, and eastern Asia, where the likelihood of multiple serotypes circulating simultaneously is reduced and therefore the human population is less likely to develop serious complications[19]. The human population is partitioned into those susceptible to infection, $H_S$, those infected, $H_I$, and those resistant to infection due to having recovered, $H_R$[146]. It is assumed that the size of the human population is constant with the exception of the entry of infected humans into the population. Human population density is estimated using the Gridded Population of the World, Version (GPWv4) estimates for 2015[147]. *Ae. albopictus* is an opportunistic feeder, and, in the absence of humans, will bite other vertebrate species. These bites cannot propagate dengue virus (outside of the geographically limited sylvatic transmission cycle[148]), but do allow for egg production and so can maintain the mosquito population. To represent this a population of non-human food sources is included in the model, $H_B$, which is estimated from field observations of rates of anthropophagy and a meta-analysis of mammal densities[149,150].

Mosquitoes are assumed to bite at a temperature-dependent rate of $b(t)$ bites per mosquito per day that is inversely proportional to the length of the gonotrophic cycle such that $b(t) = 1/G(T(t))$[43]. The proportion of uninfected mosquitoes that become infected after biting an infected human is denoted $h_\upsilon(t)$, a function that takes the form used in Liu-Helmersson et al. (2014)[151]. After the extrinsic incubation period, $\tau_{EIP}(t)$, the proportion of bites that successfully transmit an infection from an infected mosquito to an uninfected human is denoted $\upsilon_h(t)$[43,151]. After the intrinsic incubation period, $\tau_{IIP}$, the infected human is capable of transmitting the infection to new mosquitoes and recovers from the infection after the recovery period, $\tau_{REC}$. Note that these delays implicitly account for the latent period between exposure and infection without the need for the explicit inclusion of an exposed class. In this model, large mosquitoes are assumed to have no direct advantage in the act of transmission over small mosquitoes and do not bite more aggressively nor transmit disease more competently. The only advantage large mosquitoes have over small is in their extended longevity, which allows a greater proportion of mosquitoes contracting an infection to survive to pass that infection on, especially under stressful environmental conditions.

As the population model predicts the number of mosquitoes arising from a single habitat and we now wish to predict the disease transmission dynamics for a human population, it is necessary to reconcile both populations onto the same spatial scale. Mark-recapture studies have demonstrated that adult female *Ae.*

*albopictus* can disperse up to a kilometre from their release site[152]. Further, analysis of dengue epidemics has shown that disease dynamics can best be understood in $2 \text{ km} \times 2 \text{ km}$ grids[153]. For these reasons the model is simulated at a spatial scale of $4 \text{ km}^2$, using the number of humans per $4 \text{ km}^2$ for the human population in combination with an estimate of the number of larval habitats per $4 \text{ km}^2$ for the mosquito population which is halved to represent that only female mosquitoes bite. The number of productive larval habitats per $4 \text{ km}^2$, $\kappa$, is assumed to be fixed between locations and within seasons and estimated from field, surveys of larval habitats in Emilia-Romagna, Italy, Hawai'i, USA, and Maryland, USA[25,154–156].

The structured human population is represented by

$$\frac{dH_S(t)}{dt} = -R_H(t) \tag{41}$$

$$\frac{dH_I(t)}{dt} = R_H(t - \tau_{IIP}) - R_H(t - \tau_{IIP} - \tau_{REC}) + C_I(t) - C_I(t - \tau_{REC}) \tag{42}$$

$$\frac{dH_R(t)}{dt} = R_H(t - \tau_{IIP} - \tau_{REC}) + C_I(t - \tau_{REC}) \tag{43}$$

where

$$R_H(t) = \upsilon_h(t) b(t) 2\kappa \sum_{j=1}^{m} I_j \frac{H_S(t)}{H_T(t)} \tag{44}$$

and $H_T(t) = H_S(t) + H_I(t) + H_R(t) + H_B$ and $C_I(t)$ is an impulse defined as in the mosquito population model that initiates the transmission cycle through the introduction of infectious humans at a predetermined time that varies from outbreak to outbreak. Note that the inclusion of non-human food sources introduces a dilution effect, as when $H_B$ is large in relation to $H_S(t)$ the value of $R_H(t)$ will be low. This means that when the human population size is small, and therefore most bloodmeals are assumed to come from hosts that cannot become infected with dengue virus, the model predicts that only a low number of total bites by infected mosquitoes result in susceptible humans becoming infected. The omission of an explicit state-equation for humans who have been exposed to infection, but who are not yet infectious is a consequence of the intrinsic incubation period being implicitly accounted for in the delay structure through the inclusion of the term $\tau_{IIP}$.

To represent infections in the mosquito population, infectious adult classes are added, one for each environmental class, denoted $I_j(t)$. The number of mosquitoes in environmental class $j$ that become infected at time $t$ is denoted $R_{I_j}(t)$, and defined by

$$R_{I_j}(t) = h_\upsilon(t) b(t) A_j(t) \frac{H_I(t)}{H_T(t)}, \tag{45}$$

and the number of mosquitoes in environmental class $j$ that become infectious at time $t$ is denoted $M_{A_j}(t)$ and defined by

$$M_{A_j}(t) = \frac{g_{EIP}(t)}{g_{EIP}(t - \tau_{EIP}(t))} R_{I_j}(t - \tau_{EIP}(t)) S_{EIP_j}(t), \tag{46}$$

where $S_{EIP_j}(t)$ is the proportion of individuals in environmental class $j$ that survive the infectious period.

The dynamics of infectious mosquitoes can therefore be described by

$$\frac{dI_j(t)}{dt} = M_{A_j}(t) - \delta_{I_j}(t) I_j(t), \text{ for } j \in 1, \dots, m \tag{47}$$

$$\frac{d\tau_{\text{EIP}}(t)}{dt} = 1 - \frac{g_{\text{EIP}}(t)}{g_{\text{EIP}}(t - \tau_{\text{EIP}}(t))} \qquad (48)$$

$$\frac{dS_{\text{EIP}_j}(t)}{dt} = S_{\text{EIP}_j}(t)\left(\frac{g_{\text{EIP}}(t)\delta_{A_j}(t - \tau_{\text{EIP}}(t))}{g_{\text{EIP}}(t - \tau_{\text{EIP}}(t))} - \delta_{A_j}(t)\right), \quad \text{for } j \in 1, ..., m, \qquad (49)$$

where $g_{\text{EIP}}(t)$ is defined as in Brady et al.[43]. The equation for the rate of change of adults in environmental class $j$ is modified to include a term representing mosquitoes becoming exposed to infected humans and thus leaving the uninfected adult classes such that

$$\frac{dA_j(t)}{dt} = R_{A_j}(t) - M_{A_j}(t) - \delta_{A_j}(t)A_j(t), \quad \text{for } j \in 1, ..., m. \qquad (50)$$

Where $M_{A_j}(t)$ describes the rate that adults become infective. The mosquito dynamics of the SEIR model are therefore given by

$$\frac{dE_\gamma(t)}{dt} = R_{E_\gamma}(t) - M_{E_\gamma}(t) - \delta_{E_\gamma}(t)E_\gamma(t), \qquad (51)$$

$$\frac{dE_D(t)}{dt} = R_{E_D}(t) - M_{E_D}(t) - \delta_{E_D}(t)E_D(t), \qquad (52)$$

$$\frac{dE_Q(t)}{dt} = R_{E_Q}(t) - M_{E_Q}(t) - \delta_{E_Q}(t)E_Q(t), \qquad (53)$$

$$\frac{dL(t)}{dt} = R_L(t) - M_L(t) - \delta_L(t)L(t), \qquad (54)$$

$$\frac{dA_j(t)}{dt} = R_{A_j}(t) - M_{A_j}(t) - \delta_{A_j}(t)A_j(t), \quad \text{for } j \in 1, ..., m \qquad (55)$$

$$\frac{dI_j(t)}{dt} = M_{A_j}(t) - \delta_{I_j}(t)I_j(t), \quad \text{for } j \in 1, ..., m. \qquad (56)$$

The variable delay terms satisfy

$$\frac{d\tau_{E_\gamma}(t)}{dt} = 1 - \frac{g_{E_\gamma}(t)}{g_{E_\gamma}(t - \tau_{E_\gamma}(t))}, \qquad (57)$$

$$\frac{d\tau_L(t)}{dt} = 1 - \frac{g_L(t)}{g_L(t - \tau_L(t))}, \qquad (58)$$

$$\frac{d\tau_P(t)}{dt} = 1 - \frac{g_P(t)}{g_P(t - \tau_P(t))}, \qquad (59)$$

$$\frac{d\tau_{\text{EIP}}(t)}{dt} = 1 - \frac{g_{\text{EIP}}(t)}{g_{\text{EIP}}(t - \tau_{\text{EIP}}(t))}. \qquad (60)$$

The through-stage survival terms satisfy

$$\frac{dS_{E_\gamma}(t)}{dt} = S_{E_\gamma}(t)\left(\frac{g_{E_\gamma}(t)\delta_{E_\gamma}(t - \tau_{E_\gamma}(t))}{g_{E_\gamma}(t - \tau_{E_\gamma}(t))} - \delta_{E_\gamma}(t)\right), \qquad (61)$$

$$\frac{dS_L(t)}{dt} = S_L(t)\left(\frac{g_L(t)\delta_L(t - \tau_L(t))}{g_L(t - \tau_L(t))} - \delta_L(t)\right), \qquad (62)$$

$$\frac{dS_P(t)}{dt} = S_P(t)\left(\frac{g_P(t)\delta_P(t - \tau_P(t))}{g_P(t - \tau_P(t))} - \delta_P(t)\right), \qquad (63)$$

$$\frac{dS_{\text{EIP}_j}(t)}{dt} = S_{\text{EIP}_j}(t)\left(\frac{g_{\text{EIP}}(t)\delta_{A_j}(t - \tau_{\text{EIP}}(t))}{g_{\text{EIP}}(t - \tau_{\text{EIP}}(t))} - \delta_{A_j}(t)\right), \quad \text{for } j \in 1, ..., m. \qquad (64)$$

The initial conditions are the same as those for the disease free mosquito model, with the additional history $H_I(0) = H_R(0) = 0$ for $t \le 0$ and $H_S(0)$ is the initial human population density as estimated from the Gridded Population of the World, Version (GPWv4) for 2015[147]. The mosquito population is assumed to be initially entirely uninfected and mosquitoes are introduced to the system as described in Equation (32). Refer to the table in Supplementary Note 6 for an overview of all parameters and notation.

### $R_t$ derivation
To quantify how disease risk changes between regions we derive an expression for the case reproduction number $R_t$ for the phenotypic stage-structured model[157]. The case reproduction number is distinct from the basic reproduction number as it reflects how transmission risk varies as the outbreak progresses, accounting for how changes in the infection structure of the population alter the potential for the outbreak to continue as time passes[158]. We assume that a single infected human is introduced precisely at the end of the intrinsic incubation period and so can be bitten for the full duration of the infectious period. We note that for a mosquito to become infectious at time $t$ the infecting bite must have occurred at time $t - \tau_{\text{EIP}}(t)$. Therefore, the number of mosquitoes that become infectious at time $t$ per day per 4 km² can be expressed as

$$\sum_{j=1}^{m} \frac{\frac{g_{\text{EIP}}(s)}{g_{\text{EIP}}(s - \tau_{\text{EIP}}(s))}b(t - \tau_{\text{EIP}}(t))h_v(t - \tau_{\text{EIP}}(t))2\kappa A_j(t - \tau_{\text{EIP}}(t))S_{\text{EIP}_j}(t)}{H_T}. \qquad (65)$$

The infected human is assumed to remain infectious for $\tau_{\text{REC}}$ days, and so the total number mosquitoes per 4 km² that become infected as a result of the introduction of an infected human at time $t - \tau_{\text{EIP}}(t)$ is the case reproduction number for human to mosquito transmission and can be expressed as

$$\sum_{j=1}^{m} \int_t^{t+\tau_{\text{REC}}} \frac{\frac{g_{\text{EIP}}(s)}{g_{\text{EIP}}(s - \tau_{\text{EIP}}(s))}b(s - \tau_{\text{EIP}}(s))h_v(s - \tau_{\text{EIP}}(s))2\kappa A_j(s - \tau_{\text{EIP}}(s))S_{\text{EIP}_j}(s)}{H_T}ds. \qquad (66)$$

The number of susceptible humans that a single infectious mosquito bites per day at time $t$ can be expressed as

$$\frac{b(t)v_h(t)H_S(t)}{H_T}, \qquad (67)$$

and we can therefore approximate the mosquito to human case reproduction number, the total number of new human infections caused by a mosquito that became infectious at time $t$, by

$$\int_t^{t+1/\delta_{A_j}(t)} \frac{b(s)v_h(s)H_S(s)}{H_T}ds. \qquad (68)$$

Note that this is only an approximation as the lifespan of the infectious mosquito, $1/\delta_{A_j}(t)$, is determined by the environment at time $t$ and does not account for any temperature-induced changes in adult mortality that occur between times $t$ and $t + 1/\delta_{A_j}(t)$.

The human-to-human case reproduction number is the square root of the product of the human-to-mosquito and mosquito-to-human case reproduction numbers[157]. Hence, the number of new infections caused by the introduction of a single infectious human at time $t - \tau_{EIP}(t)$ can be approximated by

$$
R_{t-\tau_{EIP}(t)} = \left[ \sum_{j=1}^{m} \left( \int_{t}^{t+\tau_{REC}} \frac{\frac{g_{EIP}(s)}{g_{EIP}(s-\tau_{EIP}(s))} b(s-\tau_{EIP}(s)) h_v(s-\tau_{EIP}(s)) 2\kappa A_j(s-\tau_{EIP}(s)) S_{EIP_j}(s)}{H_T} \right. \right.
$$
$$
\left. \left. \times \left( \int_{s}^{s+1/\delta_{A_j}(s)} \frac{b(u) v_h(u) H_s(u)}{H_T} du \right) ds \right) \right]^{\frac{1}{2}}.
$$
$$(69)$$

From Equation (69), we can compute $R_{t'}$, where $t' = t - \tau_{EIP}(t)$, directly from the numerical output. We note that in Equation (69) the mosquito-to-human case reproduction number appears inside the integral for the human-to-mosquito case reproduction number. This accounts for the fact that depending on the human population and environmental conditions at the time a mosquito becomes infected, the resultant number of mosquito-to-human cases will vary. We test the validity of this approximation in two ways. In Supplementary Note 4 we demonstrate that for a population held at constant temperature when $R_t = 1 - \epsilon$ case numbers decrease and that when $R_t = 1 + \epsilon$ case numbers increase for a suitable $\epsilon$. Further, Supplementary Fig. 70 shows the average total annual number of dengue cases over the same spatial region and temporal extent that is considered in Fig. 4.

### Reporting summary

Further information on research design is available in the Nature Portfolio Reporting Summary linked to this article.

## Data availability

The ERA5-land climate reanalysis dataset can be accessed from the Copernicus Climate Data Store [https://cds.climate.copernicus.eu/]. The Gridded Population of the Word, Version (DPWv4) can be accessed through the Socioeconomic Data and Applications Center [https://sedac.ciesin.columbia.edu/]. Source data are provided as a Source Data file. The access information for the digitised life-history data and observations of field populations are provided in the Source Data file and also in the associated GitHub repository [https://github.com/DomBrass/Aedes_DDE]. Source data are provided with this paper.

## Code availability

The code used to produce these results is available in the associated GitHub repository [https://github.com/DomBrass/Aedes_DDE].

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

## Acknowledgements

D.P.B. was funded by the NERC Centre for Doctoral Training in Quantitative and Modelling skills in Ecology and Evolution (QMEE), grant number NE/P012345/1, which was supervised by S.M.W., A.C., C.A.C., B.V.P. and D.E. D.E. was supported by the Rural and Environment Science and Analytical Services Division (RESAS).

## Author contributions

D.P.B. wrote the manuscript. D.P.B., S.M.W., C.A.C., D.E. and B.V.P. designed research. D.P.B., S.M.W., C.A.C., D.E., B.V.P. and A.C. guided research and proofread the manuscript.

## Competing interests

The authors declare no competing interests.
