## [Peer Review File · Nature Communications]

Role of vector phenotypic plasticity in disease transmission as illustrated by the spread of dengue virus by *Aedes albopictus*REVIEWER COMMENTS

Reviewer #1 (Remarks to the Author):

The authors intuitively make the case for temporal effects of the environment on phenotypic plasticity. Their observations on the shortcomings of previous models are very astute – I would describe their model as being much more biologically-informed than existing models, and hence intuitively more likely to describe real-world nuances which are missed by more basic analyses. These hypotheses seem to pan out when validating their model against field data. However, I do have some criticisms of the data they have used which will require some additional systematic analyses and in some cases describing the unavoidable gaps in the data. I focus mainly on validation of the model and data utilization in my comments as I believe these to be the weakest parts of the work. Overall though, and assuming my concerns can be adequately addressed, I believe this to be an excellent and very substantial advancement in disease risk prediction and commend the authors.

114-129 and figure 1: I am not sufficiently familiar with the individual field observations and data produced. What the authors show is excellent consistence between their model and the field data from the listed papers. However, there lacks a convincing described methodology to show me that these studies are representative/exhaustive. I am certainly not arguing that the authors have cherry-picked only the studies whose data align with their predictions, but the authors do not do a sufficient job in convincing beyond reasonable doubt that this is not the case. I would like to see a paragraph detailing essentially a mini-systematic review where they have unbiasedly identified all papers with relevant data, and then systematically compared their predictions to all of them, rather than just the displayed sub-set. And this should be broken down by the type of data in fig 1 (i.e. oviposition activity, number of larvae, number of adults...). From what is written, it could be that this is an exhaustive list, a representative list, or just a cherry-picked list of neatly aligning data – convince the reader that this is one of the first two, not the latter.

137-149 and figure 2: Same comment as for figure 1 really. There have been a lot of dengue outbreaks, why are these four specifically chosen and represented here? Presumably you can only include albopictus-driven outbreaks, so this may reduce the number considerably, but there must be more than these presented. Again, I'd like to see a systematic methodology of selection of the examples, and ideally a comparison of model to outbreak for all useable outbreaks (I'm aware this may be too many to be feasible, in which case there needs to be a very convincing 'representative vs cherry-picked' explanation). And the same for vector control activities, in Guangzhou 2013-2014, you very neatly show the vector control coincides with a 'missing' observed dengue peak that your model predicts, is this an exception or does this pattern follow for a large proportion of 'missing' observed peaks – for this I'm fully aware that data on activities may be lacking, so a discussion paragraph here would suffice.

General statement – you assess the model predictions of outbreaks by comparing to real-world albopictus-driven outbreaks, by necessity of course. However, might this lead to a bias as albo-only regions may be different in terms of climate/land use to albo + aegypti regions (e.g. you have two small island examples). Not much can be done about this, but I suggest you comment on the effect on the utility and validation of your model. How can we know it is generalizable to aegypti endemic regions and e.g. mid-continental tropical regions?

Figure 4 – this figure gives me more confidence that the earlier studies were representative rather than being cherry-picked. But again the full analysis/validation is limited to Europe where there is albo but no aegypti. Again I understand the need for this, but Europe is not a representative example of general albo-driven DENV transmission – the model could be working very well here, but less well in e.g. more tropical regions (where there is more DENV). Are there enough data/resolution and land area to do these analyses for La Reunion for example, it seems strange that it is not included here? Same for Japan, I would presume there is sufficient data there – even though it's largely sub-tropical.

I would like to see some discussion of the lack of data availability and the effect on the model. E.g.

Figure 6 showing reaction norms for eggs vs temperature is a bit data-lacking, a couple of additional points could drastically change the shape of your curves. This is far more of a problem for Figure 8, especially B... Of course you have to work with the data that are available, so not a criticism of your work, but these shortcomings could have large effects on the model, especially at the higher end of the temperature range (incidentally, where you have fewer field-data for validation as noted before... This should be specifically addressed in your discussion).

Minor

Paragraph lines 59-70 (and a few later examples such as line 176) – strictly, it is 'dengue virus' which is vectored by *Ae. albopictus*, not 'dengue', which refers to the disease itself in humans.

(Remarks on code availability):

Beyond my skillset

Reviewer #2 (Remarks to the Author):

In this manuscript, the authors use a SIR transmission model with mosquito population dynamics to investigate the impact of phenotypic plasticity in vector traits on dengue transmission in the context of *Aedes albopictus*. The mosquito population dynamics sub-model is more complex than those typically encountered in the epidemic modelling literature and has been introduced in a previous work. An important feature of the present model is its ability to yield a realistic dynamics of mosquito wing-length, which the authors use as a proxy for mosquito size and lifespan. This trait varies over time due to climatic and ecological factors, resulting in subtle dynamical changes in epidemic risk. Overall, I believe that the manuscript is well written and that the implications from this model are potentially very important. I particularly appreciated several of the considerations offered in the text. However, I believe that the implications for epidemic modelling are not sufficiently articulated in the text. Please find my detailed comments below:

- The authors focus on the case of dengue and *Aedes albopictus*. Not being an expert in mosquitos, I find it unclear whether similar reasonings apply to other host-pathogen systems as well. Is phenotypic plasticity equally variable/relevant to epidemic spread in other mosquito species? I would thus suggest the authors to comment on the relevance of their findings to other systems. Eventually, the title could be made more specific as in its current form, the manuscript focuses heavily on a single application and it remains unclear how these result generalise.
- The model appears to be accurate at reproducing mean wing length data whenever such data is available (Fig 1). However, Fig 1 and Supp. plots show that such data is often available within short time intervals. Moreover, wing-length does not vary a lot during these periods, whereas the model seems to suggest that this trait can be much more variable over a longer time window. Given that dynamical variation in wing-length underpins much of downstream results, I wonder if the authors could provide further evidence to support the "U" shape displayed by wing-length? At present, I fear that the model is extrapolating and I am not sure whether these extrapolations are justified or not.
- The authors show, rather convincingly, that different modelling assumptions yield quantitative differences in epidemic dynamics. In particular, the authors show that dengue incidence and attack rate can vary considerably depending on how heterogeneity in wing-length is accounted for (or not accounted at all). However, while I do not have any doubts that the presented model is more biologically plausible than other alternatives, I'm not entirely convinced about its advantages in epidemiological modelling (e.g. in real-time tracking, planning for vector control). The practical superiority of the full model is also difficult to assess without proper model comparison: it could well be that simpler models fit the data equally well or better, but we don't know. Here I'm not suggesting to perform a proper model selection exercise, but it would be good if the authors expanded on the implications of their eco-epidemiological model over simpler models. For example, the consideration that epidemic risk varies over time due to wing-length (and that mosquito abundance alone is not sufficient to predict transmission) is extremely interesting in my opinion. Potential avenues of discussion include implications for vector control and disease persistence during inter epidemic periods. Also, what are the consequences for R_0 estimation? If I

had to fit a simpler model to data (case data, serological data), would I get a wildly different estimate of R_0 with respect to using the full model? How would these estimates then affect what we know about disease management?

- How do estimates of suitability compare with another widely used suitability index, "Index P" (Obolski et al, Methods in Ecology and Evolution, 2019; Nakase et al, Scientific data, 2023)? Does the present index provide further insights on why Italy experienced an unprecedented dengue outbreak in 2023 (please refer to Branda et al., Dengue virus transmission in Italy: surveillance and epidemiological trends up to 2023, medRxiv)?

MINOR COMMENTS

- L114 onwards: please briefly list which traits are being predicted.
- Fig 2: make x axis labels uniform (2018-2019-...), increase line width...
- L139 why complex?
- Please increase x,y tick label size where this is too small to read.
- Methods L23: "both phenotypes": what does this refer to?
- Methods L29: "of" should be "or".
- Methods Eq.(3): perhaps a typo in "1(...)"?
- Methods L134: please add further details on how expressions like that for $\Delta(T)$ are derived.
- Methods L329: I would recommend the authors to label their model as a SEIR instead of a SIR, since it does indeed account for the intrinsic incubation period. It also feels a more appropriate labelling choice for a disease such as dengue.
- L351: "an infectious humans"
- Methods L404: do you mean $R_{t'}$?
- Methods L406: "This is accounts"
- Methods why relative humidity is not included??
- Perhaps show population dynamics of mosquitoes and compare it with other population dynamics models?
- Supplementary and Main: "Emilia-Romagna" and "Catania" are the correct names.
- Supplementary P8 "Through through"
- Supplementary P9 "decease"
- Supp Fig 7A: It is not clear to me if the y axis denotes total numbers of proportions of adults.
- Supp Fig 10 legend "Simluated"
- S. 1.1.2 "year 2017". Perhaps it should be 2012.
- Supp Fig 14: the y label is cropped out.
- S2.4: is it reasonable to assume that the population in Reunion is fully susceptible against DENV, given previous circulation of the virus in that area?
- S3.2 "iin"
- In the maps, both in the main and Supp material, it would be good to highlight the locations considered in this study.

RESPONSE TO REVIEWERS' COMMENTS

We thank the reviewers for their careful consideration of our manuscript and pertinent suggestions. In response to these we have made changes that have materially improved the clarity of the manuscript. In the response that follows the reviewer's comments are in red and our point-by-point responses are in black. Where substantial changes to the manuscript have been made we provide the altered text here, a line number where the change can be found, and additionally highlight all changed text in red in the manuscript.

Reviewer #1 (Remarks to the Author):

The authors intuitively make the case for temporal effects of the environment on phenotypic plasticity. Their observations on the shortcomings of previous models are very astute – I would describe their model as being much more biologically-informed than existing models, and hence intuitively more likely to describe real-world nuances which are missed by more basic analyses. These hypotheses seem to pan out when validating their model against field data. However, I do have some criticisms of the data they have used which will require some additional systematic analyses and in some cases describing the unavoidable gaps in the data. I focus mainly on validation of the model and data utilization in my comments as I believe these to be the weakest parts of the work. Overall though, and assuming my concerns can be adequately addressed, I believe this to be an excellent and very substantial advancement in disease risk prediction and commend the authors.

We thank the reviewer for their kind comments and useful criticism which we have now addressed through additional clarifications and substantial new discussion paragraphs.

114-129 and figure 1: I am not sufficiently familiar with the individual field observations and data produced. What the authors show is excellent consistence between their model and the field data from the listed papers. However, there lacks a convincing described methodology to show me that these studies are representative/exhaustive. I am certainly not arguing that the authors have cherry-picked only the studies whose data align with their predictions, but the authors do not do a sufficient job in convincing beyond reasonable doubt that this is not the case. I would like to see a paragraph detailing essentially a mini-systematic review where they have unbiasedly identified all papers with relevant data, and then systematically compared their predictions to all of them, rather than just the displayed sub-set. And this should be broken down by the type of data in fig 1 (i.e. oviposition activity, number of larvae, number of adults...). From what is written, it could be that this is an exhaustive list, a representative list, or just a cherry-picked list of neatly aligning data – convince the reader that this is one of the first two, not the latter.

The list of studies included in our validation were the result of an extensive literature search and to the best of our knowledge we believe it is representative of the literature. Our only inclusion/exclusion criteria were that samples needed to be at a least a monthly temporal resolution in a location without *Aedes aegypti*. We made additional efforts to search the Japanese language literature where many initial studies on *Aedes albopictus* were conducted. Despite the importance of mosquito-borne disease, there is yet to be a well populated database for vectors that reports the data needed to validate population dynamics at the temporal resolution required. Much of the surveillance data that does exist is difficult to access, as many countries collect mosquito surveillance

data at the state level and then do not publicly release it as it is a matter of health policy. The population dynamics that are publicly available in the academic literature are often not digitised, serving as secondary parts of larger studies looking into other questions and so are only partially reported.

The studies we did find were found by initially searching google scholar and web of science for terms such as “*Aedes albopictus*” and then snowballing from these to find additional references. We generally encountered a high number of false positives for more specific search strings, such as “*Aedes albopictus* seasonal abundance” and did not ultimately use a systematic approach to gather validation data. Despite these limitations we think the work undertaken here is still valuable as there is currently no centralised global repository for vector population dynamics data (although there are ongoing efforts to change this) and so this represents the most comprehensive collection of population dynamics for *Aedes albopictus* to date.

The inclusion criteria for our validation dataset have now been made explicit in the text (L116-135).

“We extensively validate the model predictions of mosquito population, wing length, and disease dynamics by comparing them to field surveys of mosquito populations and disease outbreaks across the species range. To validate the population dynamics we use published surveillance data, including life-stage specific population density estimates and average trait data, from 41 locations, across 14 countries and 4 continents (see Supplementary Information 1 for the full set of validations). To validate the model predictions of dengue dynamics we compare them to historical dengue outbreaks, using reports from the outbreak to select a likely introduction scenario for dengue virus into the region and to define an area over which to simulate the model. Validation data was obtained from a comprehensive search of published studies that either observed the population dynamics of *Ae. albopictus* in the field or reported human cases of dengue. The literature search was conducted using a snowball procedure - a systematic search revealed a high number of false positives. Studies were included in our validation if the data was collected with at least a monthly resolution from a region where *Aedes aegypti*, a closely related vector species was absent. This exclusion criteria is necessary as *Ae. aegypti* is known to compete with *Ae. albopictus* for larval resources and is also a vector of dengue virus, factors which we do not account for here. Since substantial variation in case reporting effort through space and time is likely, our predictions of dengue transmission are not rescaled and the validations are intended to demonstrate that the model produces plausible disease dynamics (see also Supplementary Information 2). The comparisons presented in this paper represent the full set of datasets meeting the above criteria that were found in our literature search.”

Figure 1 has been additionally adjusted to show all locations from which we were able to access population dynamics data.

137-149 and figure 2: Same comment as for figure 1 really. There have been a lot of dengue outbreaks, why are these four specifically chosen and represented here? Presumably you can only include *albopictus*-driven outbreaks, so this may reduce the number considerably, but there must be more than these presented. Again, I'd like to see a systematic methodology of selection of the examples, and ideally a comparison of model to outbreak for all useable outbreaks (I'm aware this may be too many to be feasible, in which case there needs to be a very convincing 'representative vs cherry-picked' explanation).

As the reviewer notes, we can only include outbreaks in this validation where *Aedes albopictus* was the sole vector implicated. Further, as our goal was to predict the temporal dynamics of human

dengue cases, we also require that cases be reported at a sufficient temporal resolution from a well-defined region in space that does not consolidate data from several locations. The subset of outbreaks shown in Figure 2 is the totality of the data we were able to find that meets these criteria which is now stated in the text as part of our response to the previous comment (L116-135).

And the same for vector control activities, in Guangzhou 2013-2014, you very neatly show the vector control coincides with a ‘missing’ observed dengue peak that your model predicts, is this an exception or does this pattern follow for a large proportion of ‘missing’ observed peaks – for this I’m fully aware that data on activities may be lacking, so a discussion paragraph here would suffice.

Data on vector control activities is generally lacking. Guangzhou 2013-2014 is unusually well-studied (because of its remarkable magnitude) and so there is plenty of published information regarding the changes in vector control strategies in the region. Although each of the other outbreaks was also followed by an increase in vector control activities there is not enough information to attribute the “missing” peaks to this increase (although it is our suspicion that this is the case). As suggested this is now summarised in the new discussion paragraph addressing vector control (L312-331).

“When designing vector control campaigns to implement interventions such as the sterile insect technique, gene-drive or transgenic control, it is common to rely on mathematical models of vector population dynamics \citep{Ogunlade2023}. These models are used before implementing control measures to inform the required level of population suppression and the intensity of control measures necessary to achieve this target \citep{Oliva2021, Bier2022, Powell2022}. Similarly, after control measures have been implemented, mathematical models are a practical method to assess how effective an intervention has been by predicting how an outbreak would have progressed in the absence of control \citep{Bonds2012}. Without this additional analysis field-trials can only directly link the efficacy of their intervention to a reduction in mosquito abundance and not necessarily the reduction in human cases of disease that they aim to achieve, two factors we show here are not necessarily directly linked \citep{Balatsos2024,Zhang2024}. The 2014 outbreak of dengue in Guangzhou provides a well-studied example showing the potential of our approach as we and previous models are able to use mathematical models to assess the efficacy of the implemented control measures (see Figure~2) \citep{Tang2016,Lin2016}. However, for the majority of the outbreaks we considered, this sort of retrospective analysis is not possible due to a lack of detail in the reporting of the nature, timing, and intensity of control measures implemented. Further, to make reliable predictions to assist in the design of vector control it is imperative to use models that have been independently validated against real-world data from the target system \citep{Brady2016}. Extensive and independent validation of mathematical models of systems of VBDs is currently not standard, despite the ready availability of surveillance data and is a critical step in ensuring these predictions are robust.”

General statement – you assess the model predictions of outbreaks by comparing to real-world *albopictus*-driven outbreaks, by necessity of course. However, might this lead to a bias as *albo*-only regions may be different in terms of climate/land use to *albo* + *aegypti* regions (e.g. you have two small island examples). Not much can be done about this, but I suggest you comment on the effect on the utility and validation of your model. How can we know it is generalizable to *aegypti* endemic regions and e.g. mid-continental tropical regions?

The restriction of the model to regions where *Aedes albopictus* is the only vector is a substantial limitation inherent in our validation approach. It is unfortunately necessary as we cannot account for

the effect that competition between *albopictus* and *aegypti* will have on both population and disease dynamics. To address this our group is well underway in the development of a complementary model for *Aedes aegypti* that uses this same approach and is producing promising initial results. Combining these two models to understand dengue transmission dynamics in regions where both vectors coexist will be a substantial challenge but one that we hope will address the reviewer's questions in the long term. However, in the absence of the model for *Aedes aegypti* there are still conclusions we can make about the model's generalisability which we now examine in a new discussion paragraph addressing model generalisability (L332-349).

"Dengue transmission frequently occurs in regions where *Ae. albopictus* co-occurs with the primary vector of dengue, *Ae. aegypti*, and with which it competes for resources in larval habitats \citep{Juliano1998}. This complicates our ability to predict the effect of trait dynamics on transmission efficacy as the relationship between adult wing length, temperature, larval density, and food availability is asymmetrically altered by the degree of interspecific competition that individuals of either species experience \citep{Juliano2004,Murrell2008,Lizuain2022}. Further, in endemic regions co-circulating serotypes and previous exposure to infection complicate transmission dynamics by introducing an immune structure to the human population \citep{Gibbons2007}. An extension of our model to these areas is possible but would additionally require the development of a comparable model for *Ae. aegypti*, to quantify the effect that interspecific competition has on the life-history of both species and then to extend the disease dynamics to consider multiple co-circulating serotypes of dengue virus. However, even in these more complex transmission scenarios the underlying interaction between environmental drivers and vector trait expression is driven by the same mechanisms of phenotypic plasticity that we explore here, as they are a fundamental feature of vector ecology. By developing our understanding of the drivers of disease dynamics that arise in regions with a single vector transmitting a single serotype of dengue virus, we are now well positioned to begin addressing the more complex dynamics underlying patterns of dengue transmission in other regions."

Figure 4 – this figure gives me more confidence that the earlier studies were representative rather than being cherry-picked. But again the full analysis/validation is limited to Europe where there is *albo* but no *aegypti*. Again I understand the need for this, but Europe is not a representative example of general *albo*-driven DENV transmission – the model could be working very well here, but less well in e.g. more tropical regions (where there is more DENV). Are there enough data/resolution and land area to do these analyses for La Reunion for example, it seems strange that it is not included here? Same for Japan, I would presume there is sufficient data there – even though it's largely sub-tropical.

It is possible to perform the analysis we do in Figure 4 because in Europe *Aedes albopictus* is present and *Aedes aegypti* is not, a limitation that we now discuss in the new discussion paragraph on limitations. Although La Reunion has both highly spatially and temporally resolved case data the ERA5 environmental data we use to drive the mosquito population dynamics is not high enough resolution to perform a similar analysis, as the whole island constitutes around 20 pixels. Although this could potentially be remedied through the inclusion of microclimatic models this is beyond the scope of what we aimed to achieve here.

We also presumed that we would be able to validate the model against Japanese dengue outbreaks. However, the only instance of autochthonous transmission we could find evidence for is the 2014 Tokyo outbreak that we have modelled (see Furuya (2015)). Similar efforts in the USA found very limited evidence autochthonous transmission of dengue with the only instances outside of the

Hawai'i outbreak we model being limited to Puerto Rico, and small outbreaks in Florida and Texas where *Aedes aegypti* was the vector implicated.

I would like to see some discussion of the lack of data availability and the effect on the model. E.g. Figure 6 showing reaction norms for eggs vs temperature is a bit data-lacking, a couple of additional points could drastically change the shape of your curves. This is far more of a problem for Figure 8, especially B... Of course you have to work with the data that are available, so not a criticism of your work, but these shortcomings could have large effects on the model, especially at the higher end of the temperature range (incidentally, where you have fewer field-data for validation as noted before... This should be specifically addressed in your discussion).

The availability of laboratory data is now addressed in a new data gaps discussion paragraph (L289-311):

“The ability of mechanistic modelling approaches to incorporate detailed vector ecology allows them to produce generalisable predictions of disease risk. However, incorporating these details requires substantial amounts of life-history data that does not yet exist for many vectors, favouring the use of less detailed mechanistic models than we develop here \citep{Mordecai2019,Brady2014,Obolski2019}. Even for well-studied species such as *Ae. albopictus*, there are data gaps in our current knowledge. Some of these gaps are life-stage specific, for example, when rearing adults for infection trials it is common to record the combined life-history traits of both larval and pupal life-stages using a single generic class for juveniles, data which less detailed mechanistic approaches can readily utilise but that which we cannot. Although there are a sufficient number of studies exploring the effects of competition on larval life-history to produce well parametrised larval reaction norms, there is very little available experimental data describing the life-history of pupae which are rarely specifically reared for study. As each life-stage may be affected differently to the same environmental stressor, life-stage specific gaps in our knowledge of vector life-history potentially mean that environmental limitations on these life-stages are overlooked in our predictions of vector dynamics and consequently disease transmission dynamics. Similarly, although there has been a substantial effort to determine the lower thermal limits of the species' reaction norms due to the species northward expansion, there is a more limited amount life-history data available to describe the species responses to the extreme high temperatures which are more frequent in the regions where the majority of dengue virus transmission occurs \citep{Thomas2012,Damtew2023}. Although these limitations have not prevented our model from producing accurate predictions of population dynamics in the regions we consider, unless addressed they may limit the generalisability of our predictions of population dynamics to future climates, a factor that could be mitigated through modifications to standard experimental protocols.”

Minor

Paragraph lines 59-70 (and a few later examples such as line 176) – strictly, it is ‘dengue virus’ which is vectored by *Ae. albopictus*, not ‘dengue’, which refers to the disease itself in humans.

This is now corrected, and “dengue” and “dengue virus” are now used precisely throughout.

(Remarks on code availability):

Beyond my skillset

Reviewer #2 (Remarks to the Author):

In this manuscript, the authors use a SIR transmission model with mosquito population dynamics to investigate the impact of phenotypic plasticity in vector traits on dengue transmission in the context of *Aedes albopictus*. The mosquito population dynamics sub-model is more complex than those typically encountered in the epidemic modelling literature and has been introduced in a previous work. An important feature of the present model is its ability to yield a realistic dynamics of mosquito wing-length, which the authors use as a proxy for mosquito size and lifespan. This trait varies over time due to climatic and ecological factors, resulting in subtle dynamical changes in epidemic risk. Overall, I believe that the manuscript is well written and that the implications from this model are potentially very important. I particularly appreciated several of the considerations offered in the text. However, I believe that the implications for epidemic modelling are not sufficiently articulated in the text. Please find my detailed comments below:

- The authors focus on the case of dengue and *Aedes albopictus*. Not being an expert in mosquitos, I find it unclear whether similar reasonings apply to other host-pathogen systems as well. Is phenotypic plasticity equally variable/relevant to epidemic spread in other mosquito species? I would thus suggest the authors to comment on the relevance of their findings to other systems.

This same mechanism of phenotypic plasticity altering vector competence has been proposed in other vector species (see Cator et al. (2023)). We further discuss the generalisability of our findings to *Aedes aegypti*, the primary vector of dengue, in a new discussion paragraph. (L332-349)

“Dengue transmission frequently occurs in regions where *Ae. albopictus* co-occurs with the primary vector of dengue, *Ae. aegypti*, and with which it competes for resources in larval habitats \citep{Juliano1998}. This complicates our ability to predict the effect of trait dynamics on transmission efficacy as the relationship between adult wing length, temperature, larval density, and food availability is asymmetrically altered by the degree of interspecific competition that individuals of either species experience \citep{Juliano2004,Murrell2008,Lizuain2022}. Further, in endemic regions co-circulating serotypes and previous exposure to infection complicate transmission dynamics by introducing an immune structure to the human population \citep{Gibbons2007}. An extension of our model to these areas is possible but would additionally require the development of a comparable model for *Ae. aegypti*, to quantify the effect that interspecific competition has on the life-history of both species and then to extend the disease dynamics to consider multiple co-circulating serotypes of dengue virus. However, even in these more complex transmission scenarios the underlying interaction between environmental drivers and vector trait expression is driven by the same mechanisms of phenotypic plasticity that we explore here, as they are a fundamental feature of vector ecology. By developing our understanding of the drivers of disease dynamics that arise in regions with a single vector transmitting a single serotype of dengue virus, we are now well positioned to begin addressing the more complex dynamics underlying patterns of dengue transmission in other regions.”

Eventually, the title could be made more specific as in its current form, the manuscript focuses heavily on a single application and it remains unclear how these result generalise.

We agree that the title should be more specific and have changed the manuscript title to:

“Phenotypic plasticity in vector traits drives trends in global disease incidence: the spread of dengue virus by *Aedes albopictus*”

- The model appears to be accurate at reproducing mean wing length data whenever such data is available (Fig 1). However, Fig 1 and Supp. plots show that such data is often available within short time intervals. Moreover, wing-length does not vary a lot during these periods, whereas the model seems to suggest that this trait can be much more variable over a longer time window. Given that dynamical variation in wing-length underpins much of downstream results, I wonder if the authors could provide further evidence to support the “U” shape displayed by wing-length? At present, I fear that the model is extrapolating and I am not sure whether these extrapolations are justified or not.

In general, the relationship between high larval competition/temperatures and a reduction in wing-length is well quantified through laboratory experiments in *Aedes albopictus* and a range of other mosquito species (see for example Ezeakacha and Yee (2019)). The “U” shape is an emergent dynamical behaviour that arises from the reaction norms we have statistically fitted as a consequence of seasonal variation from low->high->low larval competition and low -> high -> low temperatures. This is generally the behaviour that is observed in field population of larvae and therefore qualitatively is as we would expect.

However, we agree with the reviewer that as this is such a critical part of our argument that we should back this up with additional field data. Unfortunately, the availability of seasonal wing-length data for *Aedes albopictus* is generally limited and the few datasets we present are the extent of the data we are aware of. However, for other mosquito species we can observe similar U-shaped trends in seasonal wing-length. For example, Foley et al. (2020) demonstrate a U-shaped wing-length dynamics in four different Anopheles vectors of malaria in Korea, the same pattern of variation is observed in various Culex species in Japan, the USA, and Australia (where the relationship becomes an “n” to match the Austral winter) , and for Aedes species in the UK. A subset of these examples is now presented in the text to provide additional context for our results. (L165-169)

“Wing length is a trait that is observable in the field where it is known to seasonally vary in a manner that qualitatively matches with the variation we have observed in populations of *Ae. albopictus* and other mosquito species (Willis1994,Packer1989,Foley2020,Kassim2013}. This variation is highly correlated with variation in adult longevity (Ezeakacha2019).”

- The authors show, rather convincingly, that different modelling assumptions yield quantitative differences in epidemic dynamics. In particular, the authors show that dengue incidence and attack rate can vary considerably depending on how heterogeneity in wing-length is accounted for (or not accounted at all). However, while I do not have any doubts that the presented model is more biologically plausible than other alternatives, I’m not entirely convinced about its advantages in epidemiological modelling (e.g. in real-time tracking, planning for vector control). The practical superiority of the full model is also difficult to assess without proper model comparison: it could well be that simpler models fit the data equally well or better, but we don’t know. Here I’m not suggesting to perform a proper model selection exercise, but it would be good if the authors expanded on the implications of their eco-epidemiological model over simpler models. For example, the consideration that epidemic risk varies over time due to wing-length (and that mosquito abundance alone is not sufficient to predict transmission) is extremely interesting in my opinion. Potential avenues of

discussion include implications for vector control and disease persistence during inter epidemic periods. Also, what are the consequences for R_0 estimation? If I had to fit a simpler model to data (case data, serological data), would I get a wildly different estimate of R_0 with respect to using the full model? How would these estimates then affect what we know about disease management?

The constant wing-length model developed in Supplementary Information 3 is the closest analogy to the simpler Ross-Macdonald style models that our framework can make. A comparison between the R_t estimates made in the full model with phenotypic plasticity and by the simpler constant wing length model variants are shown in Supplementary Figure 56, which is now mentioned in the main text (L225-226). We further show that the different model variants make different predictions of R_t which then consequently result in different dengue dynamics. As the role of environmental variation is so fundamental in driving the trait and disease dynamics it is difficult to make any more specific statements (although this is an interesting avenue for future exploration).

We directly compare the spatial distribution of risk predicted by our approach and a commonly used simpler model (Mordecai et al. 2017) in Supplementary Information 4 (referenced in the main text (L247-250)). In general our approach predicts risk in fewer places for a shorter period of time than previous approaches. This is exemplified in Europe, where we can be certain that dengue is caused by *Aedes albopictus*, and we predict risk over a more limited geographic region than the previous approach. Despite this our model predicts 38 of the 41 clusters of autochthonous dengue transmission that have occurred in Europe in the 21st century (L264-266).

Furthermore, we now discuss the use of our accurate temporal predictions of both vector and dengue dynamics, in application to vector control in a new discussion paragraph (L312-331).

“When designing vector control campaigns to implement interventions such as the sterile insect technique, gene-drive or transgenic control, it is common to rely on mathematical models of vector population dynamics \citep{Ogunlade2023}. These models are used before implementing control measures to inform the required level of population suppression and the intensity of control measures necessary to achieve this target \citep{Oliva2021, Bier2022, Powell2022}. Similarly, after control measures have been implemented, mathematical models are a practical method to assess how effective an intervention has been by predicting how an outbreak would have progressed in the absence of control \citep{Bonds2012}. Without this additional analysis field-trials can only directly link the efficacy of their intervention to a reduction in mosquito abundance and not necessarily the reduction in human cases of disease that they aim to achieve, two factors we show here are not necessarily directly linked \citep{Balatsos2024,Zhang2024}. The 2014 outbreak of dengue in Guangzhou provides a well-studied example showing the potential of our approach as we and previous models are able to use mathematical models to assess the efficacy of the implemented control measures (see Figure 2) \citep{Tang2016,Lin2016}. However, for the majority of the outbreaks we considered, this sort of retrospective analysis is not possible due to a lack of detail in the reporting of the nature, timing, and intensity of control measures implemented. Further, to make reliable predictions to assist in the design of vector control it is imperative to use models that have been independently validated against real-world data from the target system \citep{Brady2016}. Extensive and independent validation of mathematical models of systems of VBDs is currently not standard, despite the ready availability of surveillance data and is a critical step in ensuring these predictions are robust.”

- How do estimates of suitability compare with another widely used suitability index, “Index P” (Obolski et al, *Methods in Ecology and Evolution*, 2019; Nakase et al, *Scientific data*, 2023)? Does the

present index provide further insights on why Italy experienced an unprecedented dengue outbreak in 2023 (please refer to Branda et Al., Dengue virus transmission in Italy: surveillance and epidemiological trends up to 2023, medRxiv)?

Our understanding is that “Index p” is a metric that estimates the vector competence portion of a traditional R0 equation (i.e. R0 with no mosquito abundance), and so what we say about mean-field based approaches to estimating disease risk also directly applies to index P including the comparison in Supplementary Information 4 (see also comments in the main text L45-59 & 217-234).

Although 2023 is outside our study time-frame the areas of Italy where transmission was observed in 2023 correspond to areas that we consistently predict are suitable for (relatively) small dengue outbreaks (See Figure 4). Our research group is currently exploring whether or not climatic changes can explain the recent intensification of European outbreaks, but this is out with the scope of this piece of work. Our initial impression is that in Europe, although the transmission season is limited, the outbreak size is quite environmentally sensitive (see Alpes-Maritimes in Figure 2) and so it is certainly possible that 2023 was exceptionally suitable.

MINOR COMMENTS

- L114 onwards: please briefly list which traits are being predicted.

Changed to “wing-length dynamics” as this is the only trait predicted by the model that is directly validated against field-data. We do model other traits such as stage specific development rates and through-stage survival proportions but as these are not routinely measured in the field we cannot validate these predictions beyond the cumulative effect each of these have on the population’s overall dynamics. This is consistent with other environmentally sensitive VBD models (e.g., Mordecai, Caldwell, Metelmann etc).

- Fig 2: make x axis labels uniform (2018-2019-...), increase line width...

In Figure 2 the legends have been made uniform and line-width increased.

- L139 why complex?

Removed complex. As Reunion is the only location we consider where the outbreak is sustained continuously for more than a year, this outbreak is epidemiologically more complicated (resistant populations, interepidemic periods of low transmission etc.). However, we feel this is adequately conveyed by “multi-year” alone, as from the perspective of our model this outbreak is treated in the same way as all of the other outbreaks (and indeed this is the point).

- Please increase x,y tick label size where this is too small to read.

Axis labels and legend sizes have been increased throughout the text.

- Methods L23: “both phenotypes”: what does this refer to?

Changed to “both diapausing and non-diapausing eggs” to increase clarity.

- Methods L29: “of” should be “or”.

Typo corrected.

- Methods Eq.(3): perhaps a typo in “1(...)”?

We assume that hatching takes a day to complete and that only a proportion of quiescent eggs undergo this process. Therefore $h_q(t)$ is a rate but $(1 - (V-W(t)/V))$ looks initially dimensionless because there is an implicit 1/day. The 1(...) was an attempt at dimensional clarity but admittedly looks odd. We have changed the equation to $1x(...)$ to increase clarity.

- Methods L134: please add further details on how expressions like that for $\delta(T)$ are derived.

The relationship between the mortality rate from through-stage survival and stage duration is inherited from the underlying Nisbet and Gurney framework for stage-structured delay differential equations as a consequence of how they “lump” the von-Foerster equation into age classes. We feel that the argument demonstrating that this form of $\delta(T)$ is valid is too technical to reproduce here, but we now briefly explain the provenance of the equation in the text and provide a reference that supplies the full mathematical details. (L519-523)

“From the underlying Nisbet and Gurney (1983) framework we inherit a relationship between the development rate, the probability of through-stage survival and the mortality rate. We therefore use the temperature dependent reaction norms for development and through stage survival to define an expression for the temperature dependent mortality rate of active eggs, $\delta_{E_\gamma}(T) = -\log(\hat{S}_{E_\gamma}(T)g_{E_\gamma}(T))$ \citep{Nisbet1983}.”

- Methods L329: I would recommend the authors to label their model as a SEIR instead of a SIR, since it does indeed account for the intrinsic incubation period. It also feels a more appropriate labelling choice for a disease such as dengue.

We agree and the model is now referred to as an SEIR model throughout and text added to make it clear how the intrinsic incubation period is accounted for without the need for an explicit exposed class. (L748-751)

“The omission of an explicit state-equation for humans that have been exposed to infection, but who are not yet infectious is a consequence of the intrinsic incubation period being implicitly accounted for in the delay structure through the inclusion of the term τ_{IIP} .”

- L351: “an infectious humans”

This error has been corrected.

- Methods L404: do you mean $R_{t'}$?

This error has been corrected as suggested.

- Methods L406: “This is accounts”

This error has been corrected.

- Methods why relative humidity is not included??

Humidity is undoubtedly a critical factor in determining the life-span of adult mosquitoes, and so potentially critical in determining the potential of vector populations to transmit disease. However, humidity is so rarely varied in laboratory settings, that we were unable to rigorously parametrise reaction norms describing the relationship between adult longevity and humidity (see Schmidt et al. (2018)). This data gap is not unique to *Aedes albopictus* and is a barrier to considering the effects of relative humidity on the population dynamics of mosquitoes in general (Brown et al. (2023)). We are currently involved in an effort to design experimental work to obtain these parameters, but this is a substantial undertaking.

- Perhaps show population dynamics of mosquitoes and compare it with other population dynamics models?

- Supplementary and Main: “Emilia-Romagna” and “Catania” are the correct names.

Typo corrected in the main text and throughout the Supplementary Information.

- Supplementary P8 “Through through”

This error has been corrected.

- Supplementary P9 “decease”

This error has been corrected.

- Supp Fig 7A: It is not clear to me if the y axis denotes total numbers or proportions of adults.

Both Figure and legend updated for clarity, we now plot the number of infected adults per 4km².

- Supp Fig 10 legend “Simluated”

The Figure legend has been corrected.

- S. 1.1.2 “year 2017”. Perhaps it should be 2012.

This error has been corrected to 2012.

- Supp Fig 14: the y label is cropped out.

- S2.4: is it reasonable to assume that the population in Reunion is fully susceptible against DENV, given previous circulation of the virus in that area?

Although DENV had previously circulated at low levels in Reunion, the outbreak we consider here was the first large outbreak on the island since 1978 (see Vincent et al. (2023)). In these previous years DENV-2, the serotype responsible for the majority of cases in the outbreak we consider, was rarely detected (<https://www.ncbi.nlm.nih.gov/pmc/articles/PMC9333208/>). Given both of these factors, we feel it is reasonable to make the assumption that the population is fully susceptible to

DENV-2, as the proportion of the population with resistance at the start of 2017 was likely low. This is now summarised in the supplementary text (S.2.4.).

“Previous to the 2017-2021 outbreak the last large outbreak of dengue in on Reunion occurred in 1978 \citep{Vincent2023}. Further, in the years preceding this outbreak the dominant circulating serotype was DENV-I. We therefore assume that the population of Reunion is initially completely susceptible to the introduced dengue serotype.”

- S3.2 “iin”

This error has been corrected.

- In the maps, both in the main and Supp material, it would be good to highlight the locations considered in this study.

The locations of all field datasets considered are now provided in Figure 1.

REVIEWERS' COMMENTS

Reviewer #1 (Remarks to the Author):

The authors have satisfied my initial concerns with the paper. In particular, demonstrating that they had not cherry-picked examples, and the addition to the discussion regarding limitations in *Ae. albopictus*-only areas needed for validation. I recommend the manuscript for publication.

Reviewer #1 (Remarks on code availability):

This is outside of my skillset, the second reviewer seems to have covered this.

Reviewer #2 (Remarks to the Author):

I am pleased with the authors' response and I would like to commend them for the huge amount of work that went into this manuscript.